**Submission**

# The SpinParser software for pseudofermion functional renormalization group calculations on quantum magnets

Finn Lasse Buessen*

Department of Physics, University of Toronto, Toronto, Ontario M5S 1A7, Canada
Institute for Theoretical Physics, University of Cologne, 50937 Cologne, Germany
* fbuessen@physics.utoronto.ca

September 29, 2021

## Abstract

**We present the *SpinParser* open-source software [https://github.com/fbuessen/SpinParser]. The software is designed to perform pseudofermion functional renormalization group (pf-FRG) calculations for frustrated quantum magnets in two and three spatial dimensions. It aims to make such calculations readily accessible without the need to write specialized program code; instead, custom lattice graphs and microscopic spin models can be defined as plain-text input files. Underlying symmetries of the model are automatically analyzed and exploited by the numerical core written in C++ in order to optimize the performance across large-scale shared memory and/or distributed memory computing platforms.**

# 1 Introduction

Frustrated quantum magnetism presents itself as a rich platform for the study and manipulation of unusual phases of matter, including quantum spin liquid phases which evade conventional long-range order [1, 2]. Given the absence of conventional local order parameters in quantum spin liquids, their formation cannot be captured within the Landau picture of phase transitions, and their theoretical description requires novel concepts. Instead, it is understood by now that quantum spin liquids can be conceptualized within a framework of emergent gauge theories – a perspective which can efficiently capture the fractionalization of physical degrees of freedom into parton quasiparticles that exist in the background of an emergent gauge field [3, 4]. Going hand in hand with the collective constitution of new, effective quantum numbers is the formation of significant long-range entanglement, which enables the spin liquid to exhibit a variety of unusual properties, e.g., the ability to host quasiparticle excitations with non-trivial exchange statistics [5].

Detailed insight into the formation of a $Z_2$ spin liquid can be gained by the example of the Kitaev honeycomb model, where the effective low-energy theory can be explicitly constructed [6]. The manual introduction of appropriate Majorana fermionic parton operators directly exposes the underlying gauge structure and opens the door for further analytical studies of the ground state [7, 8], or even for numerical studies of the thermal ensemble at finite temperature [9, 10]. However, the exact solvability of the model breaks down when competing interactions beyond the plain Kitaev honeycomb model are included, as would be the case for most candidate material realizations [11–13]. When an analytic solution is unknown, i.e., for the vast majority of models in frustrated magnetism, it is generally challenging to make the connection from a microscopic theory to its effective low-energy description. The attempt to establish such connection poses a general conundrum not only for complicated models with a large number of competing interactions that attempt to faithfully mimic real materials, but even for minimal models like the kagome Heisenberg antiferromagnet, which has resisted a conclusive theoretical description for decades [14, 15]. In such models, answering the seemingly simple question of whether magnetic long-range order in the ground state is present or absent often requires to conduct a challenging numerical analysis. A number of numerical methods have proven themselves useful for such studies, yet all approaches have their strengths and their shortcomings.

Surely, the exact solution of the microscopic theory would be desirable, which can be achieved by an exact diagonalization (ED) of the Hamiltonian. Due to the underlying exponentially large Hilbert space, however, ED is limited to small system sizes, often yielding instructive results for two-dimensional models, but being unfeasible in three spatial dimensions. For low-dimensional systems, good results can further be obtained with the Density Matrix Renormalization Group (DMRG), which is based on an efficient (but inherently one-dimensional) matrix product state representation of wave functions [16, 17]. Similarly to ED, the approach often becomes intractable in three dimensions. Quantum Monte Carlo (QMC) simulations provide another important backbone of the numerical study of frustrated magnetism. Various adaptions of QMC can generate quasi-exact results even for three-dimensional systems, unless the computation is plagued by the fermionic sign problem, which often occurs for frustrated spin models of contemporary interest. The absence of a universally suited numerical method for the unbiased analysis of frustrated magnetism – especially in three spatial dimensions – imposes a need to further refine existing methods as well as explore novel algorithms.

Over the course of the last decade, a pseudofermion functional renormalization group (pf-FRG) approach, originally proposed by Reuther and Wölfle in 2010 [18], has established itself as a versatile technique for the analysis of ground state phase diagrams of quantum magnets. The technique is based on a recasting of spin operators in terms of complex fermions (pseudo-fermions), with the resulting strongly coupled fermion model being solved in the framework of the functional renormalization group [19]. It has been demonstrated that the pf-FRG approach

becomes exact in the large-S limit of classical spins [20], as well as the large-N limit of generalized SU(N) moments [21, 22]. The leading-order contributions of both limiting cases, which are associated with an inherent preference towards magnetic order and spin liquid ground states, respectively, are treated on equal footing within the pf-FRG, thus making it a good starting point for the unbiased analysis of competing interactions. Indeed, following its inception, it was quickly demonstrated for a number of prototypical examples of frustrated quantum magnetism that the pf-FRG approach is able to predict ground state phase diagrams which are compatible with existing data that could be obtained by means of other numerical and analytical techniques; the approach has since been successfully applied to a number of models of competing Heisenberg interactions [18, 20, 21, 23–37], including examples of long-range dipolar interactions [38, 39], as well as to models with interactions of reduced symmetry, e.g. Kitaev-like [40–46] or Dzyaloshinskii-Moriya interactions [47, 48].

The pf-FRG algorithm can be straight-forwardly applied to spin models in three spatial dimensions. In fact, many of the aforementioned studies have been conducted on three-dimensional lattice geometries. Consequently, the pf-FRG makes a large class of spin models tractable which are notoriously difficult to analyze from the perspective of many established methods that are often constrained to low-dimensional systems or small system sizes. The pf-FRG approach, in contrast, operates on an infinite representation of the underlying lattice structure with no artificial boundaries, making it suitable even for the detection of phases which are prone to form incommensurate magnetic correlations [20, 32, 46].

In this manuscript, we present the SpinParser software, which is an implementation of the pf-FRG algorithm that is designed to be easy to use and flexible in its application, giving 'out-of-the-box' access to the numerical analysis of models in quantum magnetism. The software is suited for the analysis of microscopic models comprised of two-spin interactions which preserve time-reversal symmetry. For such models, SpinParser provides access to the elastic ($\omega = 0$) component of the two-spin correlation functions. The underlying lattice geometry of the model can be flexible; the software includes an abstraction layer which automatically generates lattice representations from user-specified two-dimensional or three-dimensional lattice unit cells and automatically exploits lattice symmetries in the process. Numerically demanding calculations are enabled by the underlying C++ code with built-in support for shared-memory parallelization as well as distributed-memory parallelization in an MPI [49] environment.

The paper is structured as follows. In Sec. 2 we outline the class of models which are amenable to an analysis with the SpinParser software. In Sec. 3 we give a brief overview of the pf-FRG algorithm itself, and we discuss details of its implementation in the SpinParser software in Sec. 4. Usage instructions for the software as well as explanations on how to define custom lattice spin models are provided in Sec. 6. We round up the discussion by showing two fully worked out SpinParser calculations in Sec. 6, followed by a brief summary in Sec. 7.

## 2  Scope of the software

The *SpinParser* ('Spin Pseudofermion Algorithms for Research on Spin Ensembles via Renormalization') software is designed to solve the general class of spin models which can be captured by the microscopic Hamiltonian

$$H = \sum_{i,j} J_{ij}^{\mu\nu} S_i^\mu S_j^\nu \,, \tag{1}$$

where $J_{ij}^{\mu\nu}$ are real-valued exchange constants and the spin operator $S_i^\mu$ represents the $\mu = x, y, z$ component of a quantum spin-1/2 moment on the $i$-th lattice site. In order to solve such spin models, the software implements the pseudofermion functional renormalization group (pf-FRG) algorithm, which we briefly review in Sec. 3. For further in-depth reading on the pf-FRG approach

we refer the reader to Refs. [48, 50].

Microscopic spin models of interest can be defined on a variety of different lattice geometries. Therefore, the SpinParser software has been developed to work with generic lattice graphs that can be flexibly constructed either in two or in three spatial dimensions. The list of compatible Hamiltonians covers many influential models: the kagome Heisenberg antiferromagnet, the pyrochlore antiferromagnet, and other geometrically frustrated Heisenberg models; models with exchange frustration arising from multiple competing interactions as observed e.g. in the $J_1 J_2$-Heisenberg model on the square lattice; models with bond-directional interactions in the spirit of the Kitaev model. The list of amenable models also covers models with less symmetric spin interactions, e.g. Dzyaloshinskii-Moriya interactions or so-called $\Gamma$-interactions, which often occur alongside Kitaev interactions in real materials [11–13]. Any custom lattice spin model can be implemented, provided that it is defined on a lattice graph in which all sites are equivalent under lattice symmetry transformations and/or permutations of the spin components.

In addition to the most general implementation of the pf-FRG for the microscopic Hamiltonian Eq. (1), the software also provides more specialized implementations which are suitable for models with higher symmetry and provide increased numerical performance when applicable. One such specialized implementation is addressing models with diagonal spin interactions, which are captured by the Hamiltonian

$$H = \sum_{i,j} K_{ij}^x S_i^x S_j^x + K_{ij}^y S_i^y S_j^y + K_{ij}^z S_i^z S_j^z \,. \tag{2}$$

The exchange constants $K_{ij}^x$, $K_{ij}^y$, and $K_{ij}^z$ remain bond dependent and thus describe generalized (anisotropic) Kitaev models.

Similarly, a specialized implementation exists for generalized Heisenberg models which retain full spin-rotational symmetry. They are governed by the microscopic Hamiltonian

$$H = \sum_{i,j} J_{ij} \mathbf{S}_i \mathbf{S}_j \,, \tag{3}$$

where the interaction energy computes as the dot product spin operators $\mathbf{S}_i = \left(S_i^x, S_i^y, S_i^z\right)^T$. Due to their high symmetry, the generalized Heisenberg models can be solved most efficiently. Furthermore, within this specific implementation, the spin is not restricted to have length S=1/2; it can be generalized to arbitrary spin-S moments, following the extension of the pf-FRG proposed in Ref. [20].

In carrying out the pf-FRG analysis, the SpinParser software numerically computes the solution to a set of renormalization group flow equations which are associated with the lattice spin model under study. The flow equations describe the evolution of the (pseudofermionic) self-energy and vertex functions of the model at zero temperature and are obtained by invoking a (pseudofermionic) parton construction, see the discussion of the underlying theoretical framework as well as the involved approximations in Sec. 3. The quality of the solution is guided by a number of user-specified parameters which directly affect the numerical accuracy: the effective lattice size, the (Matsubara) frequency resolution of the vertex functions, and the precision to which the differential equations themselves are solved. Details about the numerical implementation are presented in Sec. 4.

Ultimately, the SpinParser software provides algorithms to numerically extract the two-spin correlation function of the form

$$\chi_{ij}^{\mu\nu}(i\omega) = \langle S_i^\mu(i\omega) S_j^\nu(-i\omega) \rangle \,, \tag{4}$$

with $i\omega = 0$, which – without the necessity to perform an analytic continuation – yields access to the two-spin correlation function $\chi_{ij}^{\mu\nu}(\omega = 0)$. Its Fourier transformation in momentum space,

the $\omega = 0$ component of the dynamic structure factor, is thus readily accessible and can be used to analyze the potential formation of magnetic order in the ground state. We mention that while the end point of the pf-FRG flow formally describes the zero-temperature solution of the spin model, it has been pointed out that intermediate results at finite renormalization group time can be re-interpreted as the finite-temperature solution of the spin ensemble [29]. This observation allows to assess the thermal stability of magnetically ordered ground ground states and to extract the spin-spin correlation functions at finite temperatures.

## 3 Functional renormalization group

The pseudofermion function renormalization group (pf-FRG) algorithm is a concrete implementation of the overarching general framework of the fermionic functional renormalization group [19, 51]. It can be thought of as a two-step protocol: In the first step, a microscopic spin Hamiltonian is mapped onto a (pseudo)fermionic model, which is then amenable to an analysis within the well developed framework of the fermionic FRG. More specifically, the pf-FRG approach, as originally put forward by Reuther and Wölfle [18], is built upon the mapping of spin-1/2 operators onto pseudofermions according to the rule

$$S_i^\mu = \frac{1}{2} f_{i\alpha}^\dagger \sigma_{\alpha\beta}^\mu f_{i\beta} , \tag{5}$$

where $\mu = x, y, z$ is the spin component, $\alpha, \beta = \uparrow, \downarrow$ denotes the pseudofermion spin, and summation over repeated spin indices is implicit. When subject to the local half-filling constraint $\sum_\alpha f_{i\alpha}^\dagger f_{i\alpha} = 1$, this mapping is a faithful representation of the original spin Hilbert space; the constraint is owed to the fact that the local pseudofermionic Hilbert space is four-dimensional and needs to be restricted to the physical subspace of dimension two [52]. For completeness, we mention that other types of FRG implementations for spin models also exist, but they have not yet been studied as extensively and lie outside the scope of this work. Such implementations include a formalism which is built on a mapping of spins onto Majorana fermions [53] or a formalism which avoids such mapping altogether, rendering fermionic FRG techniques inapplicable [54].

In this section, we briefly review the main aspects of the pf-FRG approach and the approximations involved. For this purpose, let us assume the general time-reversal invariant Hamiltonian for two-spin interactions as shown in Eq. (1). Transforming the spin model according to the fermionization rule Eq. (5) yields the quartic pseudofermion Hamiltonian

$$H_{\rm f} = \sum_{ij} \frac{J_{ij}^{\mu\nu}}{4} \sigma_{\alpha\beta}^\mu \sigma_{\gamma\delta}^\nu f_{i\alpha}^\dagger f_{j\gamma}^\dagger f_{j\delta} f_{i\beta} , \tag{6}$$

which is the starting point for an analysis of the spin model within the fermionic FRG framework. Note that we do not address the half-filling constraint of the fermionization rule explicitly, since any unphysical state – i.e., doubly occupied states or vacant states – leads to an effective local defect of zero spin, which is assumed to be energetically suppressed at low temperatures. An explicit construction to enforce the constraint is in principle possible by introducing an imaginary chemical potential [55], but it would dramatically lower the symmetries of the Hamiltonian and is therefore computationally unfavorable. Nonetheless, the fulfillment of the half-filling constraint can easily be checked a posteriori by verifying that adding a local interaction term $\mathbf{S}_i\mathbf{S}_i$ with a negative prefactor to the model does not alter its ground state [20, 56]. Such a term simply adds an energetic bias which locally favors non-zero spin values, i.e. the physical part of the extended pseudofermion Hilbert space.

The pseudofermion Hamiltonian in Eq. (6) is special in the sense that it does not contain a quadratic term in the pseudofermion operators. Its non-interacting (bare) propagator is therefore

simply given by the inverse Matsubara frequency $G_0(\omega) = (i\omega)^{-1}$, which results from the construction of the functional integral for the pseudofermionic theory. Upon including interactions, the full propagator would be dressed with self-energy corrections, $G(\omega) = (i\omega - \Sigma(\omega))^{-1}$, where $\Sigma(\omega)$ is the (Matsubara) frequency-dependent self-energy. While in the most general case the self-energy corrections could also depend on the spin configuration and lattice sites, it has been demonstrated that for time-reversal invariant systems of the general form as given in Eq. (1), the self-energy is diagonal in all quantum numbers and depends only on the Matsubara frequency [48]. We hence suppress lattice site and spin indices in our notation.

We now follow the well established route to set up (pseudo)fermionic FRG flow equations detailed in Refs. [48, 50, 57]. The first step of this procedure is to introduce a renormalization group cutoff $\Lambda$ to the bare propagator, which satisfies the two limiting cases

$$\begin{cases} G_0^\Lambda(\omega) \to 0 & \text{for} \quad \Lambda \to \infty \\ G_0^\Lambda(\omega) \to G_0(\omega) & \text{for} \quad \Lambda \to 0 \end{cases}, \tag{7}$$

where $G_0^\Lambda(\omega)$ denotes the cutoff-dependent bare propagator. Within the pf-FRG approach, the cutoff function is chosen to be a sharp multiplicative cutoff in the frequency dependence, yielding the cutoff-dependent bare propagator

$$G_0^\Lambda(\omega) = \frac{\theta\left(|\omega| - \Lambda\right)}{i\omega}. \tag{8}$$

Setting out from this choice for a cutoff, we can now address the formulation of FRG flow equations, which describe the change of the pseudofermionic 1-line irreducible $n$-particle interaction vertices under infinitesimal variations of the cutoff $\Lambda$. The flow equations form an exact mathematical connection between the maximally simplified model at infinite cutoff and the physically meaningful model of interest at vanishing cutoff. However, the structure of the flow equations is an infinite hierarchy of coupled integro-differential equations, which cannot be solved exactly since the flow of the $n$-particle vertex generally depends on terms up to order $n + 1$. It is therefore necessary to perform an approximation and truncate the hierarchy of differential equations at a finite order. We employ the so-called Katanin truncation [58] which yields a closed set of flow equations with terms up to $n = 2$. The truncated flow equations are an approximation for the flow of the (truncated) effective action

$$\Gamma^\Lambda\left[\bar{\phi}, \phi\right] = \sum_1 \Sigma^\Lambda(\omega_1)\bar{\phi}_1\phi_1 + \frac{1}{4}\sum_{1',2',1,2}\Gamma^\Lambda(1',2';1,2)\bar{\phi}_{1'}\bar{\phi}_{2'}\phi_2\phi_1, \tag{9}$$

which is the generating functional for the 1-line irreducible vertices via fermionic source fields $\bar{\phi}, \phi$ and contains the cutoff-dependent self-energy (one-particle vertex) $\Sigma^\Lambda(\omega_1)$ and the cutoff-dependent one-line irreducible two-particle vertex $\Gamma^\Lambda(1',2';1,2)$, where the integer indices $k$ resemble composite indices $k = (i_k, \omega_k, \alpha_k)$ of lattice site, Matsubara frequency, and spin index, respectively. The effective action can be related to the usual generating functional for $n$-point correlation functions for the purpose of calculating physical observables [18, 19, 50, 59].

The two vertex functions are computed according to the fermionic flow equations within the Katanin truncation, given by [18, 50, 60]

$$\frac{\mathrm{d}}{\mathrm{d}\Lambda}\Sigma^\Lambda(\omega_1) = -\frac{1}{\beta}\sum_2 \Gamma^\Lambda(1, 2; 1, 2)S^\Lambda(\omega_2), \tag{10}$$

where on the left hand side of the equation the dependence on the lattice site $i_1$ and spin index $\alpha_1$

is suppressed according to our notation convention, and

$$\frac{\mathrm{d}}{\mathrm{d}\Lambda}\Gamma^\Lambda(1',2';1,2) = \frac{1}{\beta}\sum_{3,4}\Big[\Gamma^\Lambda(1',2';3,4)\Gamma^\Lambda(3,4;1,2)$$
$$- \Gamma^\Lambda(1',4;1,3)\Gamma^\Lambda(3,2';4,2) - (3\leftrightarrow 4)$$
$$+ \Gamma^\Lambda(2',4;1,3)\Gamma^\Lambda(3,1';4,2) + (3\leftrightarrow 4)\Big]$$
$$\times G^\Lambda(\omega_3)S_{\mathrm{kat}}^\Lambda(\omega_4)\,, \tag{11}$$

where the prefactors $\frac{1}{\beta}$, with $\beta$ being the inverse temperature, arise from internal Matsubara summations. In the notation above, we further introduced the single-scale propagator

$$S^\Lambda(\omega) = \left(G^\Lambda(\omega)\right)^2\frac{\mathrm{d}}{\mathrm{d}\Lambda}\left(G_0^\Lambda(\omega)\right)^{-1} = \frac{\delta\left(|\omega|-\Lambda\right)}{i\omega - \Sigma^\Lambda(\omega)} \tag{12}$$

and the Katanin modified single-scale propagator

$$S_{\mathrm{kat}}^\Lambda(\omega) = -\frac{\mathrm{d}}{\mathrm{d}\Lambda}G^\Lambda(\omega) = S^\Lambda(\omega) - \left(G^\Lambda(\omega)\right)^2\frac{\mathrm{d}}{\mathrm{d}\Lambda}\Sigma^\Lambda(\omega)\,. \tag{13}$$

The initial conditions for the flow equations Eqs. (10) and (11) at infinite cutoff $\Lambda \to \infty$ are simply given by the bare interactions as specified in the pseudofermionic Hamiltonian Eq. (6) after replacing pseudofermion operators with Grassmann fields [18,50,60], and amount to

$$\Sigma^{\Lambda\to\infty}(\omega) = 0 \quad\text{and}\quad \Gamma^{\Lambda\to\infty}(1',2';1,2) = \frac{J_{i_1 i_2}^{\mu\nu}}{4}\sigma_{\alpha_{1'}\alpha_1}^\mu\sigma_{\alpha_{2'}\alpha_2}^\nu\,. \tag{14}$$

In the following discussion, and when attempting a numerical solution of the flow equations, we assume that the equations are formulated at zero temperature, i.e., the Matsubara frequencies become continuous and their summation is replaced by an integral; the prefactor of $\frac{1}{\beta}$ is replaced by $\frac{1}{2\pi}$, accordingly.

With the general set of pseudofermionic flow equations at hand, it is now of paramount importance to utilize symmetries of the underlying model in order to make it computationally tractable. It has been demonstrated in Ref. [48], that for the general time-reversal invariant Hamiltonian Eq. (1) the self-energy is a purely imaginary function which is antisymmetric in its frequency argument, i.e.,

$$\Sigma^\Lambda(\omega) \in i\mathbb{R}$$
$$\Sigma^\Lambda(\omega) = -\Sigma^\Lambda(-\omega)\,. \tag{15}$$

Furthermore, it was shown that the two-particle vertex can be efficiently parametrized by a set of 16 basis functions $\Gamma_{i_1 i_2}^{\mu\nu,\Lambda}(s,t,u)$, with $\mu,\nu = 0,\ldots,3$, as

$$\Gamma^\Lambda(1',2';1,2) = \Gamma_{i_1 i_2}^{\mu\nu,\Lambda}(s,t,u)\sigma_{\alpha_{1'}\alpha_1}^\mu\sigma_{\alpha_{2'}\alpha_2}^\nu\delta_{i_{1'}i_1}\delta_{i_{2'}i_2}\delta_{\omega_{1'}+\omega_{2'},\omega_1+\omega_2} - (1'\leftrightarrow 2')\,, \tag{16}$$

making use of the three bosonic transfer frequencies

$$s = \omega_{1'} + \omega_{2'}$$
$$t = \omega_{1'} - \omega_1$$
$$u = \omega_1' - \omega_2 \tag{17}$$

and expressing the spin dependence in the basis of Pauli matrices $\sigma^1$, $\sigma^2$, and $\sigma^3$, in conjunction with the identity matrix $\sigma^0$. Moreover, the basis functions are constrained by the symmetry relations

$$\Gamma^{\mu\nu,\Lambda}_{i_1 i_2}(s,t,u) \in \begin{cases} \mathbb{R} & \text{if} \quad \xi(\mu)\xi(\nu) = 1 \\ i\mathbb{R} & \text{if} \quad \xi(\mu)\xi(\nu) = -1 \end{cases}$$

$$\Gamma^{\mu\nu,\Lambda}_{i_1 i_2}(s,t,u) = \Gamma^{\nu\mu,\Lambda}_{i_2 i_1}(-s,t,u)$$

$$\Gamma^{\mu\nu,\Lambda}_{i_1 i_2}(s,t,u) = \xi(\mu)\xi(\nu)\Gamma^{\mu\nu,\Lambda}_{i_1 i_2}(s,-t,u)$$

$$\Gamma^{\mu\nu,\Lambda}_{i_1 i_2}(s,t,u) = \xi(\mu)\xi(\nu)\Gamma^{\nu\mu,\Lambda}_{i_2 i_1}(s,t,-u)$$

$$\Gamma^{\mu\nu,\Lambda}_{i_1 i_2}(s,t,u) = -\xi(\nu)\Gamma^{\mu\nu,\Lambda}_{i_1 i_2}(u,t,s)\,, \tag{18}$$

with the sign function

$$\xi(\mu) = \begin{cases} +1 & \text{if } \mu = 0 \\ -1 & \text{otherwise} \end{cases}. \tag{19}$$

Inserting the above parametrization into the general flow equations Eq. (10) and (11) allows us to explicitly evaluate internal summations over spin indices and contract Pauli matrices to recover a separate set of flow equations for each basis function, assuming the schematic form

$$\frac{\mathrm{d}}{\mathrm{d}\Lambda}\Sigma^\Lambda(\omega) = \dots \quad \text{and} \quad \frac{\mathrm{d}}{\mathrm{d}\Lambda}\Gamma^{\mu\nu,\Lambda}_{i_1 i_2}(s,t,u) = \dots. \tag{20}$$

Comparison of the vertex parametrization with the general pseudofermionic initial conditions, Eq. (14), shows that nine out of the 16 basis functions for the two-particle vertex may assume finite values in the limit of infinite cutoff:

$$\Sigma^{\Lambda\to\infty}(\omega) = 0 \quad \text{and} \quad \Gamma^{\mu\nu,\Lambda\to\infty}_{i_1 i_2}(s,t,u) = \frac{J^{\mu\nu}_{i_1 i_2}}{4} \tag{21}$$

for $\mu,\nu = 1,2,3 \equiv x,y,z$, and all two-particle basis functions with $\mu = 0$ or $\nu = 0$ have strictly zero initial value.

Since the resulting flow equations Eq. (20) upon inserting the full vertex parametrization are exceedingly long – e.g., expanding all two-particle vertices in the general flow equation Eq. (11) in terms of their 16 basis functions leads to 1280 terms with only few cancellations – we refer the reader to Ref. [50] for a full presentation. Here, we revert to a more instructive and slightly less explicit parametrization where we only focus on the structure of the Matsubara frequency and lattice site dependence, but do not resolve the spin indices explicitly. For this purpose, we parametrize the two-particle vertex as

$$\Gamma^\Lambda(1',2';1,2) = \Gamma^{\alpha_{1'}\alpha_{2'};\alpha_1\alpha_2,\Lambda}_{i_1 i_2}(s,t,u)\delta_{i_{1'}i_1}\delta_{i_{2'}i_2}\delta_{\omega_{1'}+\omega_{2'},\omega_1+\omega_2} - (1' \leftrightarrow 2') \tag{22}$$

and obtain the flow equation for the self-energy

$$\frac{\mathrm{d}}{\mathrm{d}\Lambda}\Sigma^\Lambda(\omega_1) = -\frac{1}{2\pi}\sum_{\alpha_2}\sum_{\omega_2}\Big[\sum_j \Gamma^{\alpha_1\alpha_2;\alpha_1\alpha_2,\Lambda}_{i_1 j}(\omega_1+\omega_2, 0, \omega_1-\omega_2)$$

$$- \Gamma^{\alpha_2\alpha_1;\alpha_1\alpha_2,\Lambda}_{i_1 i_1}(\omega_1+\omega_2, \omega_2-\omega_1, 0)\Big]S^\Lambda(\omega_2) \tag{23}$$

as well as the flow equation for the two-particle vertex

$$
\frac{\mathrm{d}}{\mathrm{d}\Lambda}\Gamma^{\alpha_{1'}\alpha_{2'};\alpha_1\alpha_2,\Lambda}_{i_1 i_2}(s,t,u) = \frac{1}{2\pi}\sum_{\alpha_3,\alpha_4}\sum_{\omega}\Big[
$$

$$
\big[G^{\Lambda}(\omega)S^{\Lambda}_{\mathrm{kat}}(s-\omega) + G^{\Lambda}(s-\omega)S^{\Lambda}_{\mathrm{kat}}(\omega)\big]
$$

$$
\times\,\Gamma^{\alpha_{1'}\alpha_{2'};\alpha_3\alpha_4,\Lambda}_{i_1 i_2}(s,\omega_{1'}-\omega,\omega-\omega_{2'})\Gamma^{\alpha_3\alpha_4;\alpha_1\alpha_2,\Lambda}_{i_1 i_2}(s,\omega-\omega_1,\omega-\omega_2)
$$

$$
+\big[G^{\Lambda}(\omega)S^{\Lambda}_{\mathrm{kat}}(\omega-t) + G^{\Lambda}(\omega-t)S^{\Lambda}_{\mathrm{kat}}(\omega)\big]
$$

$$
\times\Big[-\sum_{j}\Gamma^{\alpha_{1'}\alpha_4;\alpha_1\alpha_3,\Lambda}_{i_1 j}(\omega_1+\omega,t,\omega_{1'}-\omega)\Gamma^{\alpha_3\alpha_{2'};\alpha_4\alpha_2,\Lambda}_{j i_2}(\omega+\omega_{2'},t,\omega-\omega_2)
$$

$$
+\Gamma^{\alpha_{1'}\alpha_4;\alpha_1\alpha_3,\Lambda}_{i_1 i_2}(\omega_1+\omega,t,\omega_{1'}-\omega)\Gamma^{\alpha_{2'}\alpha_3;\alpha_4\alpha_2,\Lambda}_{i_2 i_2}(\omega_{2'}+\omega,\omega_2-\omega,-t)
$$

$$
+\Gamma^{\alpha_4\alpha_{1'};\alpha_1\alpha_3,\Lambda}_{i_1 i_1}(\omega_1+\omega,\omega-\omega_{1'},-t)\Gamma^{\alpha_3\alpha_{2'};\alpha_4\alpha_2,\Lambda}_{i_1 i_2}(\omega+\omega_{2'},t,\omega-\omega_2)\Big]
$$

$$
+\big[G^{\Lambda}(\omega)S^{\Lambda}_{\mathrm{kat}}(u+\omega) + G^{\Lambda}(u+\omega)S^{\Lambda}_{\mathrm{kat}}(\omega)\big]
$$

$$
\times\,\Gamma^{\alpha_4\alpha_{2'};\alpha_1\alpha_3,\Lambda}_{i_1 i_2}(\omega_1+\omega,\omega-\omega_{2'},u)\Gamma^{\alpha_{1'}\alpha_3;\alpha_4\alpha_2,\Lambda}_{i_1 i_2}(\omega_{1'}+\omega,\omega_2-\omega,u)\Big]. \tag{24}
$$

In order to obtain the final result, as schematically shown in Eq. (20), one would need to perform the final expansion of the two-point vertex function in its spin indices, expressing it in terms of its 16 basis functions

$$
\Gamma^{\alpha_{1'}\alpha_{2'};\alpha_1\alpha_2,\Lambda}_{i_1 i_2}(s,t,u) = \Gamma^{\mu\nu,\Lambda}_{i_1 i_2}(s,t,u)\sigma^{\mu}_{\alpha_{1'}\alpha_1}\sigma^{\nu}_{\alpha_{2'}\alpha_2}. \tag{25}
$$

Doing so generates a large number of terms in the flow equations which mix contributions between the different basis functions, but it does not alter the algebraic structure of the frequency and lattice site dependence. In their current form, therefore, the flow equations already reveal that the terms which contribute to the evolution of the two-particle vertex, cf. Eq. (24), can be grouped into three channels, each containing propagator functions which depend only on one of the three transfer frequencies $s$, $t$ or $u$. In the literature, these channels are often referred to as particle-particle scattering, particle-hole forward scattering, and particle-hole exchange scattering, respectively [19, 51]. The distinction between particle-particle and particle-hole channels thereby refers to the different relative orientations of intermediate propagator lines of the virtual states. The latter becomes more transparent in a diagrammatic representation, which is set up by identifying

$$
\Gamma^{\alpha_{1'}\alpha_{2'};\alpha_1\alpha_2,\Lambda}_{i_1 i_2}(s,t,u) \sim \quad \begin{matrix} i_2,w_2,\alpha_2 \longrightarrow i_2,w_{2'},\alpha_{2'} \\[4pt] i_1,w_1,\alpha_1 \longrightarrow i_1,w_{1'},\alpha_{1'} \end{matrix}. \tag{26}
$$

In the diagrammatic representation we implicitly assume conservation of Matsubara frequencies as well as conservation of the lattice site index along solid lines. With these conventions, the flow equations Eq. (23) and (24) are represented as (with external indices suppressed)

$$
\frac{\mathrm{d}}{\mathrm{d}\Lambda}\Sigma^{\Lambda}(\omega) = - \;\fbox{} \; + \;\fbox{} \tag{27}
$$

and

$$
\frac{\mathrm{d}}{\mathrm{d}\Lambda}\;\fbox{} = \;\fbox{}\; - \;\fbox{}\; + \;\fbox{}\; + \;\fbox{}\; + \;\fbox{}, \tag{28}
$$

where a single slashed propagator line denotes the single-scale propagator $S^{\Lambda}(\omega)$ and the slashed propagator pair should be read as $G^{\Lambda}(\omega_1)S^{\Lambda}_{\mathrm{kat}}(\omega_2) + G^{\Lambda}(\omega_2)S^{\Lambda}_{\mathrm{kat}}(\omega_1)$. The diagrammatic representation of terms is in the same order as in Eqs. (23) and (24). The first term in the flow equation for the two-particle vertex is the particle-particle scattering channel. The second, third, and fourth

terms resemble particle-hole forward scattering, which typically becomes large when the transfer frequency $t = \omega_{1'} - \omega_1$ is small. If, on the other hand, the exchange $u = \omega_{1'} - \omega_2$ is small, the particle-hole exchange scattering (last term) tends to be dominant.

In the bigger picture, when interpreting the pseudofermionic interactions in light of the original spin model they represent, special focus is on the first term of Eq. (27) and the second term of Eq. (28). Those terms involve closed loops of propagator lines, which means they imply a summation over all lattice sites. As such, they are capable of capturing long-range correlations in the lattice spin model. Indeed, it has been demonstrated that these channels are the leading order contributions in the large-$S$ limit of generalized spin-$S$ models, where magnetic order is known to prevail [20].

Attempting to solve the flow equations for a concrete spin model in general requires one to fully resolve the dependence on the spin indices by virtue of the parametrization given in Eq. (25). While the spin interactions in the most general time-reversal invariant Hamiltonian Eq. (1) lead to a large number of terms in the flow equations, its complexity can be reduced for spin models with higher symmetry. For Heisenberg models with SU(2) spin symmetry, for example, it is sufficient to consider a parametrization of the two-particle vertex under the constraints $\Gamma^{11,\Lambda}_{i_1 i_2}(s, t, u) = \Gamma^{22,\Lambda}_{i_1 i_2}(s, t, u) = \Gamma^{33,\Lambda}_{i_1 i_2}(s, t, u)$, and all basis functions with $\mu \neq \nu$ vanish [18]. Similarly, for spin models with only diagonal spin interactions, e.g. the Kitaev model or XXZ-type models, only basis functions with $\mu = \nu$ are nonzero [57]. The flow equations for all three cases – SU(2) models, Kitaev-like models, and general time-reversal invariant models – are explicitly implemented in the SpinParser code and thus allow for efficient numerical computations.

For completeness, we mention that further simplification of the two-particle vertex and its parametrization may be possible, subject to the specifics of the underlying spin interactions and the lattice geometry. Most importantly, the dependence of the basis functions $\Gamma^{\mu\nu,\Lambda}_{i_1 i_2}(s, t, u)$ on the two lattice sites $i_1$ and $i_2$ can be reduced to effectively depend only a single lattice site. To this end, we employ lattice symmetries $T$ which map the tuple $(i_1, i_2)$ onto the a transformed tuple $(i_{\text{ref}}, T(i_2))$, where $i_{\text{ref}}$ is a fixed reference site and its appearance in the flow equations can be suppressed. It is then sufficient to only compute any components of the basis functions relative to the reference site, reducing the computational cost by a factor equal to the total number of lattice sites $N_L$. Additional point group symmetries, which leave the reference site invariant, can further constrain the set of lattice sites which $i_2$ may be mapped to; the implementation of lattice symmetries is described in more details in Sec. 4.1.

Once the set of flow equations has been solved numerically, we would like to extract physical observables. To this end, the effective action Eq. (9), which is the generating functional for one-line irreducible diagrams, can be related to the generating functional of connected diagrams [59] – the essential ingredients for computing elastic two-spin correlations, i.e. the $\omega = 0$ component of the dynamic correlation function Eq. (4), of the form

$$\chi^{\mu\nu,\Lambda}_{ij} = \int_0^\beta \mathrm{d}\tau \langle S^\mu_i(\tau) S^\nu_j(0) \rangle \,, \tag{29}$$

where $\tau$ denotes the imaginary time resulting from a functional integral construction. On the order of the two-particle vertex truncation, the expression for the correlation function is given by [18]

$$\begin{aligned}
\chi^{\mu\nu,\Lambda}_{ij} = &-\frac{1}{8\pi} \int \mathrm{d}\omega_1 G^\Lambda(\omega_1)^2 \, \mathrm{tr}\,(\sigma^\mu \sigma^\nu) \\
&-\frac{1}{16\pi^2} \int \mathrm{d}\omega_1 \mathrm{d}\omega_2 G^\Lambda(\omega_1)^2 G^\Lambda(\omega_2)^2 \sigma^\mu_{\alpha_1 \alpha_{1'}} \sigma^\nu_{\alpha_2 \alpha_{2'}} \\
&\times \Gamma^\Lambda \left((i, \omega_1, \alpha_{1'}), (j, \omega_2, \alpha_{2'}); (i, \omega_1, \alpha_1), (j, \omega_2, \alpha_2)\right) \,,
\end{aligned} \tag{30}$$

where $\mu, \nu = x, y, z$ and summation over spin indices is implicit; for completeness, we also allow $\mu = \nu = 0$ in the definition, yielding the density-density correlation, with $\sigma^0$ being the identity matrix. Fourier transformation of the sum over spin-diagonal components obtains the momentum resolved spin correlations

$$\chi^\Lambda(\mathbf{k}) = \frac{1}{N_L} \sum_{i,j} \sum_{\mu=x,y,z} e^{i\mathbf{k}(\mathbf{r}_i - \mathbf{r}_j)} \chi_{ij}^{\mu\mu,\Lambda}, \tag{31}$$

where $\mathbf{r}_i$ is the position of the $i$-th lattice site and the normalization is by the number of lattice sites $N_L$ which the summations are performed over. We loosely refer to $\chi^\Lambda(\mathbf{k})$ as the structure factor, although one should keep in mind that the static structure factor (with no explicit dependence on time or frequency) in the literature is typically defined via the *equal-time* spin correlations, whereas our definition is based on the *elastic* spin correlations. The dominant magnetic ordering vector $\mathbf{k}_{\max}$, if present, can be inferred from the maximum of the structure factor, the peak susceptibility $\chi_{\max}^\Lambda = \max_{\mathbf{k}} \chi^\Lambda(\mathbf{k})$.

Note that the expression for the peak susceptibility $\chi_{\max}^\Lambda$ depends on the cutoff parameter $\Lambda$, but only the limit $\Lambda \to 0$ resembles the physical solution. In practice, however, it is imperative to trace the full evolution of the peak susceptibility as a function of $\Lambda$: Since we make use of a number of symmetries in the parametrization of the effective action (including time-reversal symmetry and – depending on the model under study – spin rotational symmetry), the flow equations are not suited to describe configurations which would break these symmetries. Consequently, whenever one studies spin models by means of the pf-FRG approach, whose transition into their low-temperature phases would imply the spontaneous breaking of symmetries, one typically observes a breakdown of the smooth RG flow, which manifests as a divergence or a kink in $\Lambda$-dependence of the spin correlations [18, 19]. This behavior is qualitatively different from the one in parameter regimes in which the ground state of the spin model preserves all symmetries; in the latter case, the $\Lambda$-dependence of the spin correlations remains smooth. The difference between the two allows us to map out phase diagrams with respect to magnetically ordered ground states and symmetry-preserving quantum spin liquid ground states.

We illustrate the foregoing discussion of the flow breakdown with an example. Consider a spin model of nearest-neighbor Heisenberg interactions on the kagome lattice, which is governed by the Hamiltonian

$$H = J \sum_{\langle i,j \rangle} \mathbf{S}_i \mathbf{S}_j, \tag{32}$$

where the sum runs over nearest neighbor pairs of lattice sites $i$ and $j$. Two decisively different scenarios are possible, depending on the choice of the interaction constant $J$. If we choose antiferromagnetic interactions, $J > 0$, the model becomes the kagome Heisenberg antiferromagnet, a paradigmatic model of frustrated quantum magnetism which leads to a spin liquid ground state. As such, the flow of the peak susceptibility is expected to remain smooth down to lowest cutoff, see Fig. 1a. Conversely, if we choose $J < 0$, the resulting model is a simple ferromagnet, which harbors a ground state with broken spin-rotational symmetry. As a manifestation of the broken symmetry, we observe a kink in the flow of the peak susceptibility at a finite critical RG scale $\Lambda_c$, see Fig. 1b. While the solution of the flow equations for $\Lambda < \Lambda_c$ is unphysical due to the occurrence of the breakdown, denying us exploration of the ordered phase itself, we can inspect the structure factor just above the critical scale $\Lambda_c$, where the solution of the flow equations is still valid. Already at this finite RG scale we observe the buildup of dominant correlations at the Brillouin zone center, see the inset of Fig. 1b, which indicates incipient ferromagnetic order. In this manner, facilitating the pf-FRG approach, it is possible to explore the magnetic ordering tendencies and structure factors for a plethora of models in quantum magnetism.

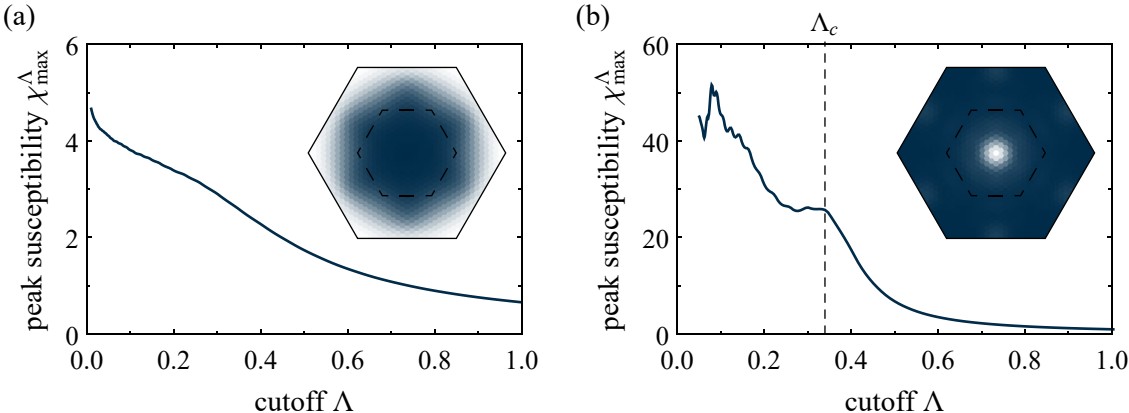

Figure 1: **Renormalization group flow** of the magnetic correlations for (a) the kagome Heisenberg antiferromagnet and (b) the kagome Heisenberg ferromagnet. The antiferromagnet shows a smooth evolution of the susceptibility down to lowest cutoff, indicative of the absence of spontaneous symmetry breaking. In the ferromagnetic example, the flow displays a kink (the *breakdown* of the smooth flow) near $\Lambda_c \approx 0.35$, which signals the onset of ferromagnetic order. The insets show the structure factor across the extended Brillouin zone, with the dashed lines denoting the first Brillouin zone, plotted at (a) $\Lambda = 0$ and (b) $\Lambda = \Lambda_c$. The color code is separately normalized for each panel, ranging from low intensity (dark blue) to high intensity (white).

## 4 Implementation details

In the previous section, we have outlined the concept of the pf-FRG algorithm and its application to a general class of quantum spin models, which are captured by the microscopic Hamiltonian Eq. (1). In this section, we provide details about the specific implementation of the pf-FRG algorithm in the SpinParser software. Aspects of the implementation, which affect the numerical performance and precision of the computation, can be broadly summarized into three groups: (i) The vertex functions $\Sigma(\omega)$ and $\Gamma_{i_1 i_2}^{\mu\nu,\Lambda}(s,t,u)$, as defined by Eqs. (15) and (18), depend on lattice site indices, which are defined on an infinite lattice graph; similarly, the frequency arguments are continuous and can assume unbounded values. The dependence of the vertex functions on the lattice site and frequency arguments must therefore be restricted to a finite set of numbers. (ii) The solution of the flow equations is performed numerically, implying that the cutoff parameter $\Lambda$ in the underlying differential equations Eq. (20) can only be incremented by finite amounts, and its discretization impacts the numerical precision of the solution. (iii) Large-scale calculations require an efficient parallelization scheme of the calculations, which needs to be devised and implemented. We comment on aspects of all three groups individually in the following subsections.

### 4.1 Lattice truncation and symmetry analysis

Microscopic quantum spin models are typically defined on an infinite lattice graph. The single-particle vertex function $\Sigma(\omega)$, which implicitly depends also on the suppressed lattice site index $i_1$ and spin index $\alpha_1$, was shown to be independent of the lattice site index – i.e., for its calculation, we can simply fix the index $i_1$ to an (arbitrary) reference site, say, $i_{\text{ref}}$ [48]. Unfortunately, the situation is more complicated for the two-particle vertex function $\Gamma_{i_1 i_2}^{\mu\nu,\Lambda}(s,t,u)$, whose dependence on two lattice site indices is more intricate. However, in analogy to the single-particle vertex function where we removed the dependence on one lattice site index by fixing a reference site, we argue in the following that the dependence of the two-particle vertex function on two lattice sites can effectively be reduced to depend only on a single lattice site, which may further be constrained by additional lattice symmetries.

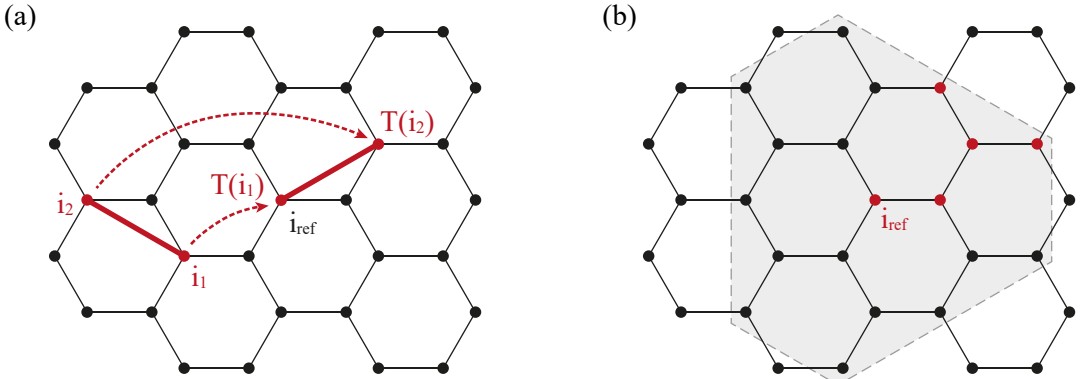

Figure 2: **Lattice truncation** and symmetry relations. (a) Lattice symmetry transformation $T$, which maps the two lattice sites $i_1$ and $i_2$ to a fixed reference site $i_{\text{ref}} = T(i_1)$ and $T(i_2)$, respectively. (b) Truncation of a two-particle vertex function. The lattice truncation around lattice site $i_{\text{ref}}$ is illustrated by the gray shaded area for truncation range $L = 3$. Within the truncation range, which contains $N_s^{\text{total}} = 19$ lattice sites, vertices need to be parametrized only with respect to the $N_s = 5$ lattice sites colored in red, since they form the (symmetry-)irreducible basis set for the lattice site dependence of the two-particle vertex functions.

We shall begin by fixing a reference site $i_{\text{ref}}$ in the lattice. Further, we shall assume that all sites in the lattice are equivalent[1] (and the SpinParser software is, in fact, only applicable to lattices for which this assumption is true.) That assumption implies that for any lattice site $i_1$ in the lattice, there exists a transformation $T$ which maps $i_1$ to our reference site $i_{\text{ref}}$. For lattices with a monatomic basis, such transformations would simply be translations by a multiple of the primitive lattice vectors. For lattices with a nontrivial basis, however, the transformations become more complicated and may involve rotations or mirror operations, since they need to provide mappings between the different basis sites (which, typically, are not simple translations.) While lattice symmetries in other contexts are often straightforwardly discussed as the mapping of one lattice site to another, here we are interested in the simultaneous action of a lattice symmetry transformation on a pair of lattice sites: Given that the transformation $T$ maps $i_1$ onto the reference site $i_{\text{ref}}$, i.e. $T(i_1) = i_{\text{ref}}$, we also require knowledge about its action on a second lattice site $i_2$ – which is going to be mapped to $T(i_2)$. Such transformation of a two-site object is illustrated in Fig. 2a. On the level of the basis functions of the two-particle vertex, it allows us to establish the mapping

$$\Gamma_{i_1 i_2}^{\mu\nu,\Lambda}(s,t,u) \mapsto \Gamma_{i_{\text{ref}} T(i_2)}^{\mu\nu,\Lambda}(s,t,u), \tag{33}$$

where the two vertex values must be equivalent by symmetry, see also the discussions in Refs. [48, 50]. Since we can fix $i_{\text{ref}}$ arbitrarily, the set of all transformations $T$ (allowing to map any site $i_1$ onto the reference site) effectively reduces the dependence of the vertex basis functions on two lattice sites to just a single lattice site index.

Next, we need to reduce the dependence of the vertex function $\Gamma_{i_{\text{ref}} T(i_2)}^{\mu\nu,\Lambda}(s,t,u)$ on the a priori infinite set of lattice sites $T(i_2)$ to a finite set, which is numerically tractable. To this end, we perform a truncation of the vertex function: If the distance between the lattice sites $i_{\text{ref}}$ and $T(i_2)$ is greater than a certain truncation range $L$, we set the vertex value to zero, i.e.,

$$\Gamma_{i_{\text{ref}} T(i_2)}^{\mu\nu,\Lambda}(s,t,u) = \begin{cases} \Gamma_{i_{\text{ref}} T(i_2)}^{\mu\nu,\Lambda}(s,t,u) & \text{if } \|T(i_2), i_{\text{ref}}\|_b \leq L \\ 0 & \text{else} \end{cases}, \tag{34}$$

---

[1]More generally, every site in the lattice spin model must be equivalent under joint lattice transformations and permutation of spin components.

where $\|\cdot, \cdot\|_b$ is the norm which measures the distance of two lattice sites by the minimal number of lattice bonds it takes to connect the sites. Thereby we guarantee that all finite vertex functions are spanned by a finite set of $N_s^{\text{total}}$ lattice sites within the truncation range, which we can represent numerically. We emphasize that this vertex truncation does not resemble a calculation on a finite lattice (neither with open nor with periodic boundary conditions), since it does not introduce an artificial boundary to the system. Rather, it can be interpreted analogously to a series expansion in the lattice site: Upon increasing the truncation range, the precision of the calculation is systematically increased and eventually the result converges to the thermodynamic limit [50]. Since the underlying lattice geometry itself always remains genuinely infinite, this approach is suitable also for spin models with ground states of incommensurate magnetic order [20, 32, 46].

But we can restrict the representation of vertex functions even further. For a given pair of lattice sites $i_1$ and $i_2$, there may exist multiple transformations $T_1, \ldots, T_n$ which map $i_1 \mapsto i_{\text{ref}}$ but have different $T_1(i_2) \neq \ldots \neq T_n(i_2)$. In other words, there exist point group transformations $U$ which leave $i_{\text{ref}}$ invariant, and which can be exploited to define an irreducible set $U(T(i_2))$, for any $i_2$ within the truncation range, which spans the minimal number of vertex basis functions, see Fig. 2b. These lattice transformations are subject to some constraints: Not all lattice spin models necessarily preserve the full symmetry of the underlying lattice. The interaction terms in the Kitaev honeycomb model [6], for example, break the three-fold rotation symmetry of the underlying lattice. Such interactions are common in many models of current interest, so in order to exploit an even larger class of symmetries, we lift the point group transformation $U$ to act on the product space of lattice site indices and spin indices, i.e., in addition to performing a lattice transformation, we may also perform a spin transformation. For an efficient numerical implementation, we restrict the spin transformation to be a global permutation of the three spin components $x$, $y$, and $z$. This yields the final symmetry relation

$$\Gamma_{i_1 i_2}^{\mu\nu,\Lambda}(s, t, u) = \Gamma_{i_{\text{ref}} U(T(i_2))}^{U(\mu)U(\nu),\Lambda}(s, t, u), \tag{35}$$

where $U$ is chosen such that the number of lattice sites $N_s$ in the image set of $U \circ T$ for any lattice sites $i_1$ and $i_2$ is as small as possible. It is thus sufficient to numerically parametrize the vertex functions only over the $N_s$ lattice sites which span the irreducible image of $U \circ T$ and obtain all remaining components of the vertex functions via the symmetry relation above. However, identifying the full set of symmetry transformations requires a lot of work, and they are custom tailored to the specific choice of the lattice spin model.

The SpinParser software automatically performs the search for symmetry transformations to minimize $N_s$ and it parametrizes the vertex basis functions accordingly. The search algorithm for lattice symmetries relies on an internal real-space representation of the lattice, which is constructed up to an absolute precision of $\varepsilon = 10^{-5}$; for the symmetries to be detected correctly, it is thus necessary to define the lattice geometry (primitive lattice vectors and basis site positions) at a precision of $\varepsilon$ or higher (cf. Sec. 5.2). In order to achieve good performance at runtime, the symmetry calculations are performed only once at the beginning of the code execution and the results are then tabulated for later use throughout the solution of the flow equations. In particular, we also tabulate lattice sites $U(T(j))$ (and associated symmetry transformations) on which products of two vertex functions of the form $\sum_j \Gamma_{i_1 j} \Gamma_{j i_2}$ (spin indices and frequency arguments suppressed) assume finite values, i.e. $j$ lies within the truncation range around both $i_1$ and $i_2$. Such lattice summations appear in the flow equations for the two-particle vertex function, cf. Eq. (24), and make up a significant share of the computational workload.

With the tabulation of the abovementioned lattice summations in place, the computational complexity of the expression scales only with the number of irreducible lattice sites $N_s$, instead of the total number of lattice sites within the truncation range, $N_s^{\text{total}}$. In order to demonstrate this numerically, we simulate a Heisenberg antiferromagnet (with the energy scale of the interaction constant set to $J = 1$) on different lattice geometries with various truncation ranges $L = 3, \ldots, 12$.

(a)

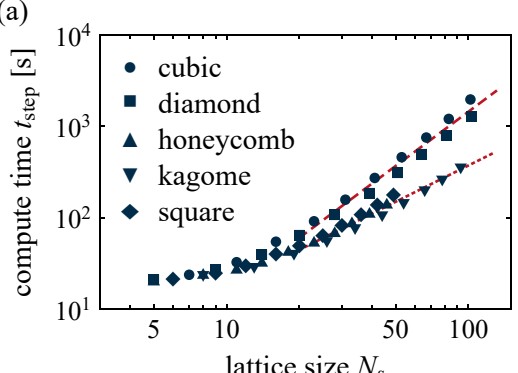

(b)

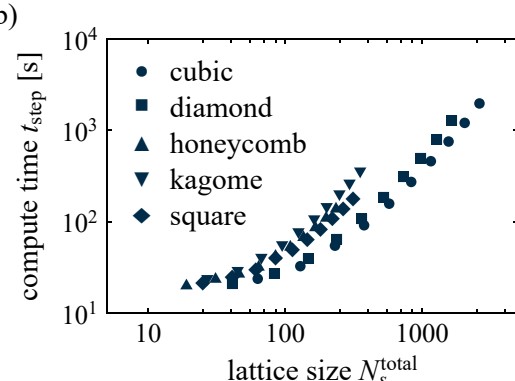

Figure 3: **Lattice geometry** and computational complexity. The compute time $t_{\text{step}}$ is measured for a single cutoff step in the solution of the flow equations on various lattice geometries (three-dimensional cubic and diamond lattice, as well as two-dimensional honeycomb, kagome, and square lattice). For each lattice, different truncation ranges $L = 3, \ldots, 12$ are considered[2]. Timing is measured on Intel Xeon Phi 7250-F Knights Landing processors, utilizing 8 cores per simulation. (a) Scaling as a function of the number of parametrized sites $t_{\text{step}} \sim N_s^\alpha$ shows deviations between two-dimensional ($\alpha = 1.32$, dotted line) and three-dimensional ($\alpha = 1.98$, dashed line) lattices. (b) Plotted as a function of the total number of lattice sites within the truncation range $N_s^{\text{total}}$, slight deviations between the scaling behavior of the different lattice geometries become visible, depending on the degree of symmetry inherent to the lattice.

As shown in Fig. 3a, for three-dimensional lattice geometries (cubic lattice and diamond lattice) the scaling of the computing time is approximately $t_{\text{step}} \sim N_s^\alpha$, with $\alpha = 1.98$. For the two-dimensional lattices (honeycomb lattice, kagome lattice, and square lattice) and the investigated truncation ranges, the scaling appears softer, $\alpha = 1.32$. However, it is expected that for larger truncation ranges, the scaling eventually approaches $\alpha = 2$: For two-dimensional lattices, the aforementioned tabulated lattice summations $\sum_j \Gamma_{i_1 j} \Gamma_{j i_2}$ typically contain fewer terms than their three-dimensional analogues; this implies that subleading terms, which scale as $\alpha = 1$, have more relative weight and the leading-order scaling $\alpha = 2$ is only observed at larger overall lattice sizes.

For comparison, we plot the same benchmark calculations as a function of the total number of lattice sites $N_s^{\text{total}}$ within the truncation range, see Fig. 3b. While the number of parametrized lattice sites only goes up to $N_s = 103$ (diamond lattice at $L = 12$) for the truncation ranges considered here, the total number of lattice sites is up to $N_s^{\text{total}} = 2625$ (cubic lattice at $L = 12$), highlighting the great simplification of the computational problem achieved by exploiting lattice symmetries. At the same time, the plot visualizes the dependence of the computational complexity on the details of the underlying lattice graph: For a fixed number of lattice sites $N_s^{\text{total}}$, the actual computing time $t_{\text{step}}$ can vary significantly between the different lattices, depending on the degree of symmetry in the lattice.

---

[2]For the benchmark, we consider nearest-neighbor Heisenberg models with interaction constant $J = 1.0$ on various lattice geometries and truncation ranges. We employ the numerical core optimized for SU(2) symmetric spin models. Frequencies are discretized logarithmically between $\omega = 0.005$ and $\omega = 50.0$ with $N_\omega = 64$. The cutoff is initialized at $\Lambda = 50.0$ and subsequently integrated down to $0.1$ with a multiplicative step size of $0.95$. Timing is recorded and averaged over the next 5 integration steps for $\Lambda < 0.1$. Note that for some non-frustrated lattice geometries, this cutoff may lie below a critical $\Lambda_c$, such that the calculation loses its physical meaning; the technical benchmark, however, remains valid.

## 4.2 Frequency discretization

One key ingredient to solving the pf-FRG flow equations is the numerical treatment of the underlying frequency structure. The single-particle vertex $\Sigma^\Lambda(\omega)$ is parametrized by a single frequency $\omega$, and each basis function $\Gamma_{i_1 i_2}^{\mu\nu,\Lambda}(s,t,u)$ of the two-particle vertex is parametrized by a set of three bosonic transfer frequencies $s$, $t$, and $u$. While the flow equations are formally derived at zero temperature, where frequency dependence is a continuous quantity, their numerical solution – in practice – can only be performed on a discrete support space of a finite number of frequency values. We therefore define a discrete mesh of $N_\omega$ positive frequency points $\omega_1, \ldots, \omega_{N_\omega}$, which are typically chosen logarithmically dense around zero (but in principle can be chosen arbitrarily, see Sec. 5.1.) The full (discretized) frequency space is then spanned symmetrically around zero by the $N_\omega^{\text{total}} = 2N_\omega$ supporting mesh points $-\omega_{N_\omega}, \ldots, -\omega_1, \omega_1, \ldots, \omega_{N_\omega}$. It would now be straightforward to define the single-particle vertex function $\Sigma^\Lambda(\omega)$ by assigning to every frequency value $\omega$ the function value $\Sigma^\Lambda(\omega_n)$ at the mesh point $\omega_n$ closest to $\omega$; and by assigning to every tuple of transfer frequencies $(s,t,u)$ the value of two-particle vertex basis functions $\Gamma_{i_1 i_2}^{\mu\nu,\Lambda}(\omega_{n_s}, \omega_{n_t}, \omega_{n_u})$ at the nearest mesh points $(\omega_{n_s}, \omega_{n_t}, \omega_{n_u})$. However, in the following, we shall formulate a refined scheme in order to reduce the computational complexity as well as reduce the numerical error which results from the discretization procedure.

First, we exploit the symmetry relation Eq. (15), which defines anti-symmetry of the single-particle vertex function $\Sigma^\Lambda(\omega)$ in its frequency argument $\omega$. With this symmetry transformation in place, it is sufficient to model the vertex function only in the positive frequency half-space with $\omega > 0$; all remaining function values at negative frequency values are read off from their symmetry equivalents. Next, we need to specify the procedure to retrieve the vertex function at arbitrary positive frequency values $\omega$, which do not necessarily coincide with one of the discrete frequency mesh points. Any vertex value is therefore obtained within a linear interpolation scheme on the discrete frequency mesh. To this end, for any frequency value $\omega$, after mapping it onto the positive half-space, we determine the nearest lesser discrete mesh point $\omega_<$ and the nearest greater frequency point $\omega_>$. The interpolated vertex function is then calculated as

$$\Sigma^\Lambda(\omega) = \frac{\omega_> - \omega}{\omega_> - \omega_<}\Sigma^\Lambda(\omega_<) + \frac{\omega - \omega_<}{\omega_> - \omega_<}\Sigma^\Lambda(\omega_>). \tag{36}$$

In case the desired frequency point, at which the vertex function is evaluated, is lesser (greater) than the minimum (maximum) discrete mesh point, the vertex is extrapolated as a constant value which coincides with the function value at the minimum (maximum) discrete mesh point.

The two-particle vertex basis functions $\Gamma_{i_1 i_2}^{\mu\nu,\Lambda}(s,t,u)$, which depend on three independent frequency arguments, are treated in close analogy: We only parametrize the basis functions on the approximately $\frac{N_\omega^3}{2}$ frequency points in the positive octant with $s \geq 0$, $t \geq 0$, $u \geq 0$, and $s \geq u$, since all remaining function values can be obtained by invoking the symmetry relations Eq. (18), which separately guarantee (anti-)symmetry in each of the three transfer frequencies, as well as an exchange relation between the two transfer frequencies $s$ and $u$. Within this parametrized octant, function values for arbitrary transfer frequency tuples $(s,t,u)$ are obtained by linear interpolation between the nearest lesser discrete frequency points $(s_<, t_<, u_<)$ in every dimension and the nearest greater frequency points $(s_>, t_>, u_>)$, respectively. The interpolation in three-dimensional

(a)

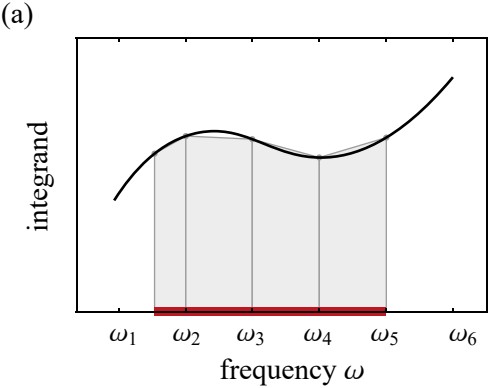

(b)

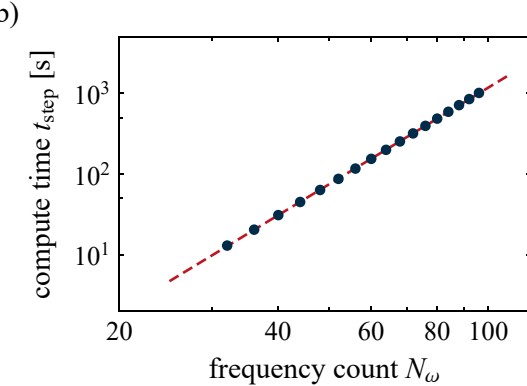

Figure 4: **Frequency integration** and computational complexity. (a) Frequency integrals in the flow equations are approximated by a trapezoidal integration scheme, where over a given integration domain (indicated in red) the integrand is approximated by trapezoidal segments (gray shaded areas). (b) The compute time for a single cutoff step in the solution of the flow equations scales as $t_{\text{step}} \sim N_\omega^\alpha$, with $\alpha = 3.98$ (dashed line). Data shown is on the kagome lattice with truncation range $L = 10$ for a Heisenberg antiferromagnet[3]. The timing is measured on Intel Xeon Phi 7250-F Knights Landing processors, utilizing 8 cores per simulation.

frequency space is performed as

$$
\begin{aligned}
\Gamma_{i_1 i_2}^{\mu\nu,\Lambda}(s,t,u) = \frac{u_> - u}{u_> - u_<} & \left[ \frac{t_> - t}{t_> - t_<} \left[ \frac{s_> - s}{s_> - s_<} \Gamma_{i_1 i_2}^{\mu\nu,\Lambda}(s_<,t_<,u_<) + \frac{s - s_<}{s_> - s_<} \Gamma_{i_1 i_2}^{\mu\nu,\Lambda}(s_>,t_<,u_<) \right] \right. \\
& \left. + \frac{t - t_<}{t_> - t_<} \left[ \frac{s_> - s}{s_> - s_<} \Gamma_{i_1 i_2}^{\mu\nu,\Lambda}(s_<,t_>,u_<) + \frac{s - s_<}{s_> - s_<} \Gamma_{i_1 i_2}^{\mu\nu,\Lambda}(s_>,t_>,u_<) \right] \right] \\
+ \frac{u - u_<}{u_> - u_<} & \left[ \frac{t_> - t}{t_> - t_<} \left[ \frac{s_> - s}{s_> - s_<} \Gamma_{i_1 i_2}^{\mu\nu,\Lambda}(s_<,t_<,u_>) + \frac{s - s_<}{s_> - s_<} \Gamma_{i_1 i_2}^{\mu\nu,\Lambda}(s_>,t_<,u_>) \right] \right. \\
& \left. + \frac{t - t_<}{t_> - t_<} \left[ \frac{s_> - s}{s_> - s_<} \Gamma_{i_1 i_2}^{\mu\nu,\Lambda}(s_<,t_>,u_>) + \frac{s - s_<}{s_> - s_<} \Gamma_{i_1 i_2}^{\mu\nu,\Lambda}(s_>,t_>,u_>) \right] \right] .
\end{aligned}
\tag{37}
$$

Similar to the treatment of the single-particle vertex, we perform a constant extrapolation in every dimension if a transfer frequency lies outside the region spanned by the discrete frequency mesh.

Typically, throughout the solution of the pf-FRG flow equations, we do not only access the vertex functions at isolated frequency points. Rather, – especially in the calculation of the two-particle vertex function – we need to perform one-dimensional line integrals embedded in the three-dimensional frequency parameter space, cf. Eq. (24). When these integrals of the form $\int f(\omega) \mathrm{d}\omega$ are evaluated numerically for an arbitrary integrand function $f(\omega)$, further approximations are required; in the SpinParser code, we employ a trapezoidal integration scheme. That is, $f(\omega)$ is evaluated at a sequence of discrete points $\omega_1, \omega_2, \ldots$ and the integral is approximated by a sum over the trapezoids spanned by the integrand values $f(\omega_1), f(\omega_2), \ldots$, see Fig. 4a. The discrete points $\omega_1, \omega_2, \ldots$ are chose to coincide with the frequency mesh points $\omega_1, \ldots, \omega_{N_\omega}$ on

---

[3]For the benchmark, we consider a nearest-neighbor Heisenberg model with interaction constant $J = 1.0$ on the kagome lattice with truncation range $L = 10$ and employ the numerical core optimized for SU(2) symmetric spin models. Frequencies are discretized logarithmically between $\omega = 0.005$ and $\omega = 50.0$ with $N_\omega$ between 32 and 96. The cutoff is initialized at $\Lambda = 50.0$ and subsequently integrated down to 0.1 with a multiplicative step size of 0.95. Timing is recorded and averaged over the next 5 integration steps for $\Lambda < 0.1$.

which the vertex functions are defined. In this way, increasing the overall number of frequency points $N_\omega$ coherently increases the numerical precision of the calculation [50].

Since the integrand typically is a complicated expression which involves multiple evaluations of the vertex functions (cf. the flow equation for the two-particle vertex Eq. (24)), it is crucial that the frequency interpolations of the vertex functions are performed efficiently. For reaching a satisfactory performance in accessing two-particle vertex values, we note that throughout the solution of the flow equations, it is often necessary to retrieve vertex values $\Gamma_{i_1 i_2}^{\mu\nu,\Lambda}(s, t, u)$ of different basis components $\mu, \nu$ and lattice sites $i_1, i_2$ (especially when an internal lattice summation is performed, see Sec. 4.1) for a constant set of transfer frequency arguments $s$, $t$, and $u$. In this situation, a large share of the work associated with a vertex interpolation only needs to be performed once, and the interpolation weights

$$\frac{s_> - s}{s_> - s_<} \, , \, \frac{s - s_<}{s_> - s_<} \, , \, \frac{t_> - t}{t_> - t_<} \, , \, \frac{t - t_<}{t_> - t_<} \, , \, \frac{u_> - u}{u_> - u_<} \, , \, \frac{u - u_<}{u_> - u_<} \tag{38}$$

as well as the mesh frequencies

$$s_< \, , \, s_> \, , \, t_< \, , \, t_> \, , \, u_< \, , \, u_> \tag{39}$$

can be buffered for future use. In fact, we do not buffer the mesh frequencies themselves, but rather their position in terms of a linear memory offset, which allows for even faster access of the associated vertex values. Aspects of the memory layout of the vertex functions are further discussed in Sec. 4.4.

The invocation of symmetry relations in combination with buffered vertex interpolation grants us a huge speedup in computing time. However, the algorithmic scaling of the computational complexity remains steep, and it is expected that the computing time scales as $t_{\text{step}} \sim N_\omega^\alpha$ with $\alpha \approx 4$. The scaling exponent, on the one hand, is a consequence of the discretization of the vertex functions, where the size of the underlying frequency mesh to leading order (i.e. for the two-particle vertex function) scales as $N_\omega^3$. On the other hand, the computation of the one-dimensional frequency integrals in the flow equations for the two-particle vertex via the trapezoidal integration routine outlined earlier in this section contributes an additional scaling factor of $N_\omega$. As exemplified in Fig. 4b for a Heisenberg antiferromagnet on the kagome lattice, we numerically observe a scaling exponent of $\alpha = 3.98$, which is very close to the theoretical prediction.

## 4.3 Differential equation solver

With the lattice truncation and the vertex discretization scheme in place, the solution of the pf-FRG flow equations Eq. (20) is within reach. The initial conditions of the vertex functions at infinite cutoff $\Lambda \to \infty$ are known, see Eq. (21) – however, the true limit of infinite cutoff cannot be implemented numerically. Therefore, for a numerical solution, the vertex functions are initialized at an initial cutoff $\Lambda_i$ which is chosen to be much greater than any intrinsic energy scale of the spin system under study, and therefore closely resembles the limit of infinite cutoff. Similarly, while the true physical solution of the vertex functions would be recovered at $\Lambda = 0$, in practice it is sufficient to determine the solution at a final cutoff $\Lambda_f$, which is small compared to any intrinsic energy scale of the system. The solution of the vertex functions at the final cutoff $\Lambda_f$ is then obtained by re-integrating the flow equations as

$$\Sigma^{\Lambda_f}(\omega) = \Sigma^{\Lambda_i}(\omega) + \int_{\Lambda_i}^{\Lambda_f} \frac{\mathrm{d}}{\mathrm{d}\Lambda} \Sigma^\Lambda(\omega) \, \mathrm{d}\Lambda \tag{40}$$

and

$$\Gamma_{i_1 i_2}^{\mu\nu,\Lambda_f}(s, t, u) = \Gamma_{i_1 i_2}^{\mu\nu,\Lambda_i}(s, t, u) + \int_{\Lambda_i}^{\Lambda_f} \frac{\mathrm{d}}{\mathrm{d}\Lambda} \Gamma_{i_1 i_2}^{\mu\nu,\Lambda}(s, t, u) \, \mathrm{d}\Lambda \, . \tag{41}$$

The integrals in the equations above – and thus the solution of the coupled differential equation – are computed with the Euler method: For vertex functions which are known at some cutoff $\Lambda$, the new vertex functions at a slightly reduced cutoff $\Lambda - \delta\Lambda$ (with $\delta\Lambda$ small) are obtained by linear extrapolation. For the single-particle vertex, the extrapolation is calculated as

$$\Sigma^{\Lambda-\delta\Lambda}(\omega) = \Sigma^{\Lambda}(\omega) - \delta\Lambda \frac{\mathrm{d}}{\mathrm{d}\Lambda} \Sigma^{\Lambda}(\omega) \,, \tag{42}$$

and the two-particle vertex is obtained in a similar manner. In this spirit, the interval between the initial cutoff $\Lambda_i$ and the final cutoff $\Lambda_f$ is divided into $N_\Lambda$ discrete cutoff points which act as support for the numerical stepping towards $\Lambda_f$. Typically, the cutoff values are chosen logarithmically dense around $\Lambda_f$, but in principle any distribution of cutoff points can be defined, see Sec. 5.1. Since the number of cutoff points $N_\Lambda$ defines the number of points at which the flow equations need to be evaluated, the total computation complexity scales linearly with $N_\Lambda$, and increasing the number of cutoff points systematically reduces numerical errors [50].

## 4.4 Vertex functions and parallelization

The solution of the pf-FRG flow equations can be computationally demanding, especially for spin models with reduced symmetry [48]. It is therefore crucial to enable an efficient parallelization of the algorithm not just on shared memory compute platforms, but also across distributed memory architectures. In principle, the parallelization of the pf-FRG algorithm is simple: At every step in the integration of the flow equations, i.e., at every discrete cutoff value encountered, a large number of independent flow equations for the vertex basis functions need to be computed. With the number of (parametrized) single-particle vertex functions $\Sigma^{\Lambda}(\omega)$ equaling $N_\omega$ and the number of two-particle vertex functions $\Gamma_{i_1 i_2}^{\mu\nu,\Lambda}(s,t,u)$ being approximately $8N_\omega^3 N_s$ (a prefactor of 16 arising from the different basis components $\mu, \nu$ and a factor of $\frac{1}{2}$ from the exchange symmetry between the transfer frequencies $s$ and $u$), the typical number of independent equations to compute can be up to $\mathcal{O}(10^8)$ [32,48], putting little constraint on the maximum number of compute cores over which the workload can be parallelized. Yet, after every step in the integration of the flow equations, the results need to be synchronized across all compute nodes, which introduces some communication overhead to the parallelization. In the following, we discuss how the parallelization is implemented in SpinParser in an attempt to reduce the communication overhead.

Due to the inherent simplicity of the single-particle vertex function $\Sigma^{\Lambda}(\omega)$ – with its sole dependence on one frequency argument, it is effectively a one-dimensional data structure – we focus our discussion on the two-particle vertex $\Gamma_{i_1 i_2}^{\mu\nu,\Lambda}(s,t,u)$, which after exploiting the lattice symmetries discussed in Sec. 4.1 is a 6-dimensional object (two basis index dimensions $\mu$ and $\nu$, three frequency dimensions $s$, $t$, and $u$, as well as one symmetry-reduced lattice site index). The vertex function is mapped onto linear memory space as follows[4]: The dependence on the two transfer frequencies $s$ and $u$ is joined to a single dimension of length $\frac{N_\omega(N_\omega+1)}{2}$, which represents all pairs $s$ and $u$ with $s >= u$; this dimension has the largest memory strides. The second dimension of length $N_\omega$ is the dependence on the transfer frequency $t$, followed by two dimensions of length 4, comprising the basis indices $\mu$ and $\nu$, respectively. The last dimension of length $N_s$ is the lattice site dependence, and it is stored contiguously in memory. The rationale behind this memory layout is that it is often required to perform frequency interpolations for a fixed set of transfer frequencies $s$, $t$, and $u$ over all combinations of basis indices $\mu$ and $\nu$, as well as over all lattice site indices. As mentioned in Sec. 4.2, it is then possible to calculate the linear interpolation

---

[4]Note that the memory structure discussed here applies to the "TRI" numerical backend. For the "SU2" backend, the memory layout is two separate memory strains for $\Gamma_{i_1 i_2}^{00,\Lambda}(s,t,u)$ and $\Gamma_{i_1 i_2}^{33,\Lambda}(s,t,u)$, with the order of the remaining frequency and lattice site dependence the same as for the "TRI" backend. Similarly, the "XYZ" numerical backend has four separate memory strains for $\Gamma_{i_1 i_2}^{00,\Lambda}(s,t,u)$, $\Gamma_{i_1 i_2}^{11,\Lambda}(s,t,u)$, $\Gamma_{i_1 i_2}^{22,\Lambda}(s,t,u)$, and $\Gamma_{i_1 i_2}^{33,\Lambda}(s,t,u)$.

weights only once, and subsequently apply them efficiently to all combinations of basis indices $\mu,\nu$ and lattice sites, which are all stored contiguously in memory.

Moreover, the two-particle vertex memory layout is beneficial for the parallelization across multiple compute nodes. The total workload (i.e. the total number of differential equations that need to be solved) is separated into blocks of $16N_s$ differential equations each, such that every block of work is associated with a set of differential equations of fixed frequency structure, but spanning all basis function indices and lattice sites – which are stored contiguously in memory and hence allow for the aforementioned efficient buffering of frequency interpolation weights separately on every compute node. In addition, the structure of the flow equations for the two-particle vertex is such that there exists one term which contains a combined frequency integral and lattice site summation; this term (cf. Eq. (24)) contributes a significant share to the computational workload. While the lattice site summation is performed at constant transfer frequency arguments $s$, $t$, and $u$, the boundaries of the frequency integral depend on the transfer frequency $t$, resulting in an augmented dependence of the overall computational workload within one block on $t$. Consequently, the memory layout was chosen such that superblocks of $16N_sN_\omega$ vertex entries, spanning all combinations of indices $t, \mu, \nu$ and lattice sites, are stored contiguously in memory and the associated differential equations can be solved with a smaller variability of computing time between different superblocks. The number of such superblocks, $\frac{N_\omega(N_\omega+1)}{2}$, – for typical parameters this is $\mathcal{O}(10^3)$ [32, 48] – is still large enough to allow for an efficient parallelization across multiple compute nodes.

Nonetheless, small variations in the expected computing time per superblock may still appear, because the computational complexity of the internal frequency integrals in the flow equations also depends on the value of the transfer frequencies $s$ and $u$, as they directly impact the size of the integration domain. The parallelization across compute nodes with distributed memory architecture is therefore equipped with a load balancing system. Schematically, the parallelization is implemented as follows (illustrated in Fig. 5). We assume that the SpinParser software is executed in an MPI environment [49] with one MPI rank per compute node; each compute node is assumed to have access to multiple shared memory CPU cores. Upon executing SpinParser, one MPI rank assumes a coordinating role, which we refer to as the *main* rank, whereas all remaining ranks will be referred to as *worker* ranks. We further assume that the initialization phase of the code has completed, i.e. all parameters for the calculation have been read (see Sec. (5.1)) and the symmetry analysis of the underlying lattice spin model (as outlined in Sec. 4.1) has been performed on every rank. In the first step of the numerical solution of the flow equations, the cutoff parameter $\Lambda$, the single-particle vertex $\Sigma^\Lambda(\omega)$, and the two-particle vertex $\Gamma_{i_1i_2}^{\mu\nu,\Lambda}(s,t,u)$ are prepared with the appropriate initial conditions on the main rank and subsequently broadcasted to all worker ranks. Next, the flow of the single-particle vertex, $\frac{\mathrm{d}}{\mathrm{d}\Lambda}\Sigma^\Lambda(\omega)$, is computed; to this end, on the main rank, the total computational workload (i.e. the flow equations for each frequency component of the vertex) is divided into units of work (sets of frequency components that need to be computed) which are then successively delegated to the worker ranks, as well as to additional compute threads on the main rank. Each rank then performs the assigned calculation and returns the result to the main rank. Once all compute ranks have completed their calculations, the resulting single-particle vertex flow $\frac{\mathrm{d}}{\mathrm{d}\Lambda}\Sigma^\Lambda(\omega)$ is broadcasted to all compute nodes, since its knowledge is required by every rank for the impending computation of the flow of the two-particle vertex. The computation of the latter, $\frac{\mathrm{d}}{\mathrm{d}\Lambda}\Gamma_{i_1i_2}^{\mu\nu,\Lambda}(s,t,u)$, is performed within the same scheme of delegating blocks of work to a set of worker ranks: The computational work is divided into units of work; for the flow of the two-particle vertex, the unit size typically typically is a multiple of the superblock size $16N_sN_\omega$, thus enabling an efficient calculation of vertex interpolations on every rank. One initial unit of work is delegated to each worker rank; note that the size of the initial unit of work is chosen such that every rank is expected to compute multiple such units. Whenever a compute rank completes its assigned work, the result is returned to the main rank. The main rank, in turn, delegates addi-

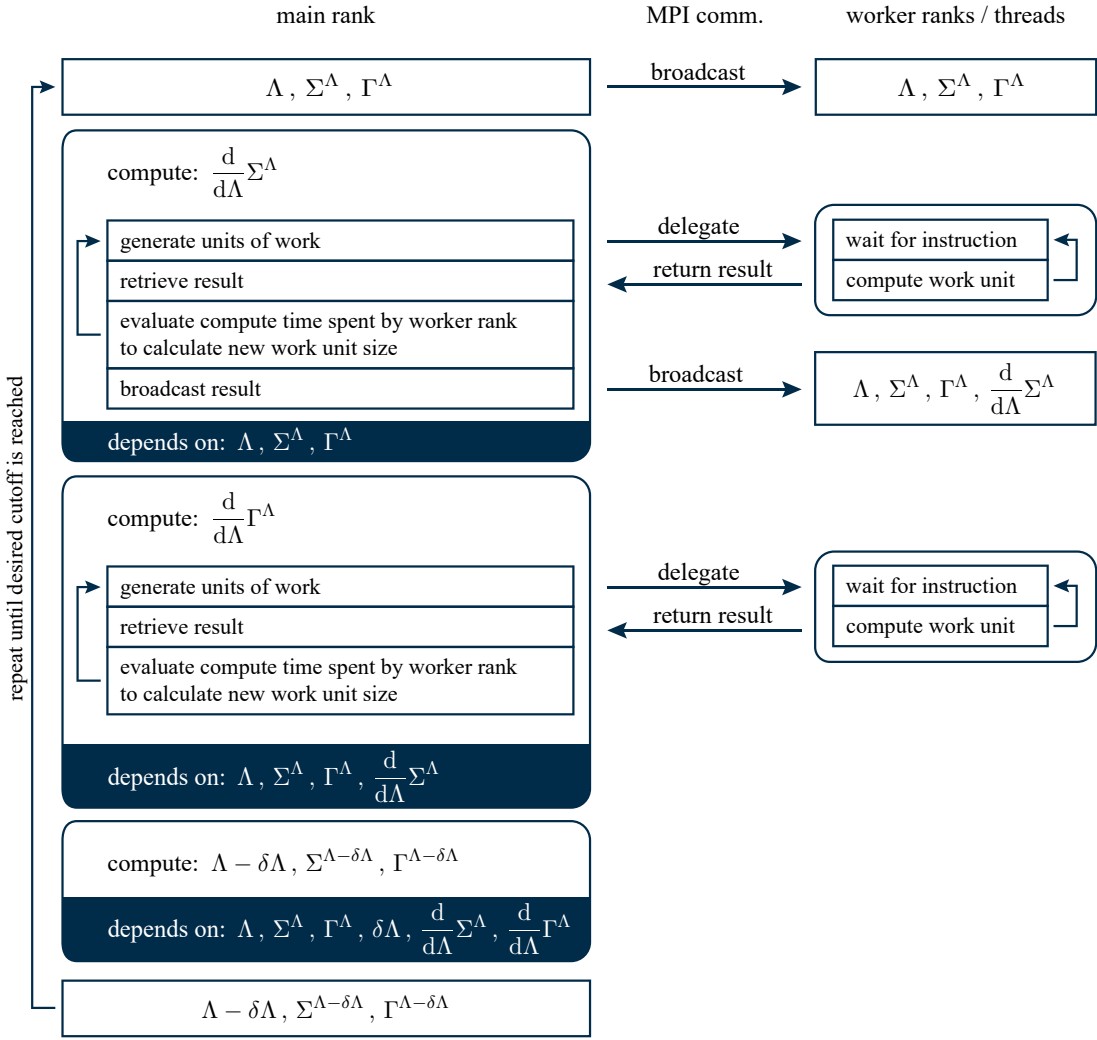

Figure 5: **Schematic algorithm** for the solution of pf-FRG flow equations. Initially, the cutoff $\Lambda$, as well as the single-particle vertex $\Sigma^\Lambda(\omega)$ and the two-particle vertex $\Gamma^{\mu\nu,\Lambda}_{i_1 i_2}(s, t, u)$ are broadcasted across all compute nodes. Note that in the figure, indices are suppressed for the sake of readability. Next, the flow $\frac{\mathrm{d}}{\mathrm{d}\Lambda}\Sigma^\Lambda(\omega)$ of the single-particle vertex is computed and the result broadcasted across all compute nodes. Subsequently, the two-particle vertex flow $\frac{\mathrm{d}}{\mathrm{d}\Lambda}\Gamma^{\mu\nu,\Lambda}_{i_1 i_2}(s, t, u)$ is computed and the result returned to the main rank. Finally, the updated vertex functions $\Sigma^{\Lambda-\delta\Lambda}(\omega)$ and $\Gamma^{\mu\nu,\Lambda-\delta\Lambda}_{i_1 i_2}(s, t, u)$ at reduced cutoff $\Lambda - \delta\Lambda$ can be calculated; the latter calculation is the only operation which is not parallelized across multiple (distributed memory) compute nodes.

(a)  (b)

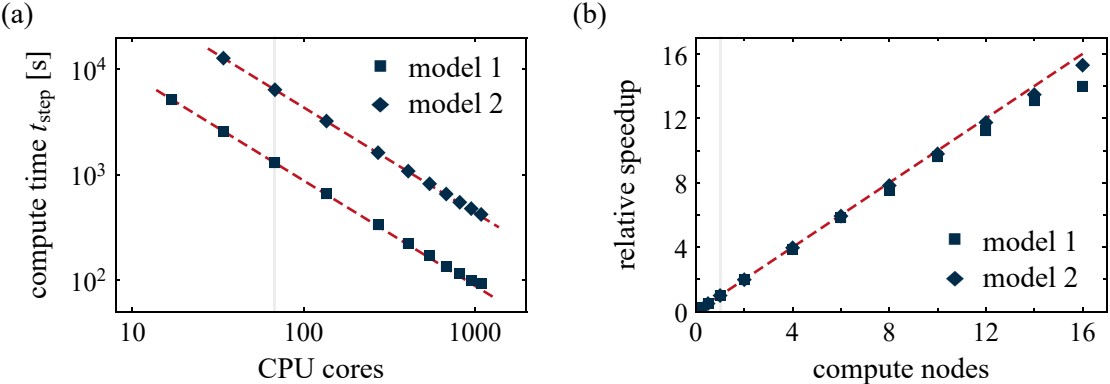

Figure 6: **Parallelization scaling** on distributed memory computing platforms. The compute time $t_{\text{step}}$ is measured for a single cutoff step in the solution of the pf-FRG flow equations for spin models of the most generic form given in Eq. (1) on a honeycomb lattice with truncation range $L = 10$. Models 1 and 2 are computed at $N_\omega = 64$ and $N_\omega = 96$, respectively[5]. Timing is measured on Intel Xeon Phi 7250-F Knights Landing processors with 68 physical CPU cores per compute node. (a) Compute time plotted as a function of the number of physical CPU cores participating in the calculation. The red dashed lines indicate perfect scaling relative to the performance on a single compute node (gray line). (b) Relative speedup with respect to the performance on a single compute node (gray line) plotted as a function of the number of compute nodes participating in the calculation. The red dashed line indicates perfect scaling.

tional units of work until the entire computation is complete. Note that the main rank keeps track of the computing time of each rank and dynamically adjusts the size of newly generated units of work such that all compute ranks are expected complete their work at approximately the same time. Finally, the extrapolation of the vertex functions by a small cutoff step $\delta\Lambda$ is performed on the main rank to obtain the new cutoff value $\Lambda - \delta\Lambda$, the single-particle vertex $\Sigma^{\Lambda-\delta\Lambda}(\omega)$, and the two-particle vertex $\Gamma_{i_1 i_2}^{\mu\nu,\Lambda-\delta\Lambda}(s,t,u)$ as described in Sec. 4.3. This entire routine (a single cutoff step in the solution of the differential equations) is repeated until the desired cutoff value is reached.

We conclude the discussion of the parallelization by benchmarking its efficiency. Varying the number of compute nodes and CPU cores utilized in the computation, we measure the compute time $t_{\text{step}}$ for a single cutoff step in the solution of the flow equations and determine its scaling relative to the performance on a single compute node. As displayed in Fig. 6a, the compute time scales almost perfectly. Small deviations become visible only when the compute time is smaller than $t_{\text{step}} \approx 100$ s (measured on Intel Xeon Phi 7250-F Knights Landing processors with 68 physical CPU cores per compute node). In the regime where $t_{\text{step}}$ remains above that threshold, the relative speedup scales approximately linearly with the number of compute nodes and the parallelization overhead remains negligible (Fig. 6b).

## 5  Usage instructions

The SpinParser software consists of a single executable named `SpinParser`, which can be run from the terminal. When running the executable, the mandatory argument `TASKFILE` needs to

---

[5]For the benchmark, we consider a nearest-neighbor Heisenberg-Kitaev-$\Gamma$ model on the honeycomb lattice with truncation range $L = 10$. Frequencies are discretized logarithmically between $\omega = 0.005$ and $\omega = 50.0$ with $N_\omega = 64$ ($N_\omega = 96$) points. The cutoff is initialized at $\Lambda = 50.0$ and subsequently integrated down to 0.1 with a multiplicative step size of 0.95. Timing is recorded and averaged over the next 5 integration steps for $\Lambda < 0.1$.

be provided, and a list of optional arguments may further be included:

```
SpinParser [OPTION]... TASKFILE
```

The mandatory argument is used to pass the file path to a so-called "task file" to the executable, in which the lattice spin model is specified along with additional parameters that are required to uniquely define the numerical problem. We describe the structure of such a task file in detail in Sec. 5.1. In addition to the parameters specified in the task file, the executable evaluates the environment variable `OMP_NUM_THREADS` in order to determine the number of threads which should be utilized for the computation. If the SpinParser executable is launched in an MPI environment, a hybrid parallelization is performed, where every MPI rank spawns the number of threads defined in `OMP_NUM_THREADS`. Furthermore, the following optional arguments can be provided to the executable:

```
-h [ --help ]
```

Print a help message which contains a list of possible arguments that may be passed to the executable, and exit. No calculation is performed.

```
-r [ --resourcePath ] DIR
```

Define a search path `DIR` to scan for resource files, which contain lattice and spin model definitions, see Secs. 5.2 and 5.3.

```
-v [ --verbose ]
```

Make output more verbose.

```
--debugLattice
```

Only construct the lattice representation and exit. No calculation is performed, but the lattice information is written to disk, see Sec. 5.4.

```
-t [ --checkpointTime ] TIME (=3600)
```

Define a time interval `TIME` in seconds at which checkpoint files are written to disk. Incomplete calculations can be resumed from these checkpoint files, see Sec. 5.1.

```
-f [ --forceRestart ]
```

Force a restart of the calculation, even if previous checkpoint files are available, see Sec. 5.1.

```
-d [ --defer ]
```

Do not perform measurements; instead, write the full vertex data to disk. Measurements are performed when the executable is run with the same task file for a second time.

## 5.1 Structure of a task file

The task file is a plain-text file, in which the computational problem is defined. The full problem specification does not only contain a description of the quantum spin model itself, i.e., the precise coupling constants $J_{ij}^{\mu\nu}$ in the general spin Hamiltonian Eq. (1) as well as the underling lattice graph, but it also contains parameters which define the numerical precision for the solution, e.g. the discretization and boundary of the frequency grid as well as the cutoff parameter values over which the flow equations are integrated. Furthermore, the task file contains a list of physical observables which should be measured throughout the computation. Task files are conveniently

```
1  <task>
2      <parameters>
3          <frequency discretization="exponential">
4              <min>0.005</min>
5              <max>50.0</max>
6              <count>32</count>
7          </frequency>
8          <cutoff discretization="exponential">
9              <min>0.1</min>
10             <max>50</max>
11             <step>0.95</step>
12         </cutoff>
13         <lattice name="square" range="4"/>
14         <model name="square-heisenberg" symmetry="SU2">
15             <j>1.0</j>
16         </model>
17     </parameters>
18     <measurements>
19         <measurement name="correlation"/>
20     </measurements>
21 </task>
```

Listing 1: **Example of a task file**. The task file contains relevant information for the discretization of the frequency dependence (lines 3–7), the discretization of the cutoff parameter (lines 8–12), the underling lattice graph of the quantum spin model (line 13), as well as the spin model itself (lines 14–16). Note that the lattice and spin model definitions are only referenced, and their actual implementation is found in separate files, see Secs. 5.2 and 5.3 for details. Finally, the desired measurements of physical observables are defined in line 19.

written in an XML structure; a complete example of a task file is shown in Lst. 1. In the remainder of this subsection, we discuss in detail the structure of the task file.

Every task file contains one `task` node on the top level (line 1 in Lst. 1), which contains the two sub nodes `parameters` and `measurements` (lines 2 and 18, respectively, in Lst. 1). The former of the sub nodes must contain one instance of the nodes `frequency`, `cutoff`, `lattice`, and `model` each, while the latter can contain any number of `measurement` nodes.

It is possible two define the discretization and the boundaries of the frequency spectrum in two different ways. The first option is to use a logarithmically spaced mesh of discrete frequencies, which is generated symmetrically around zero. Such an automatically generated frequency distribution is specified via

```
<frequency discretization="exponential">
    <min>0.005</min>
    <max>50.0</max>
    <count>32</count>
</frequency>
```

where the `min` and `max` parameters specify the boundaries $\omega_{\min}$ and $\omega_{\max}$ of the frequency mesh on the positive half-axis, and the negative frequencies are generated implicitly by symmetry. The overall number of positive frequencies $N_\omega$ is defined by the parameter `count`; the total number of positive and negative frequencies is therefore $N_\omega^{\text{total}} = 2N_\omega$. Positive frequency mesh points

$\omega_n$ for $n = 0, \ldots, N_\omega - 1$ are generated according to the distribution

$$\omega_n = \omega_{\min} \left( \frac{\omega_{\max}}{\omega_{\min}} \right)^{\frac{n}{N_\omega - 1}} . \tag{43}$$

Alternatively, the frequency mesh can be defined explicitly by listing all frequency points on the positive half-axis, where negative frequencies are again added implicitly by symmetry. An example for the explicit definition of frequencies would look as follows:

```
<frequency discretization="manual">
   <value>0.005</value>
   <value>0.0067298</value>
   <!-- any number of frequency values can be listed here -->
   <value>37.1482</value>
   <value>50.0</value>
</frequency>
```

Similarly, the discretization of the cutoff parameter $\Lambda$ needs to be specified. It can, too, either be generated automatically or be defined manually. While the automatically generated logarithmic discretization around zero is in principle the same as for the previously discussed frequency discretization (but restricted to positive values only), its specification takes slightly different arguments:

```
<cutoff discretization="exponential">
   <min>0.1</min>
   <max>50</max>
   <step>0.95</step>
</cutoff>
```

Here, instead of specifying the total number of discrete cutoff points, a multiplicative step size $b$ is provided by the parameter `step` in addition to the lower and upper boundaries $\Lambda_{\min}$ and $\Lambda_{\max}$ specified in `min` and `max`, respectively. Based on the step size, the cutoff parameter values $\Lambda_n$ for $n = 0, \ldots, \left\lfloor \log_b \left( \frac{\Lambda_{\min}}{\Lambda_{\max}} \right) \right\rfloor$ are generated as

$$\Lambda_n = \Lambda_{\max} b^n . \tag{44}$$

Alternatively, the cutoff parameter discretization can also be specified explicitly:

```
<cutoff discretization="manual">
   <value>50.0</value>
   <value>47.5</value>
   <!-- any number of cutoff values can be listed here -->
   <value>0.106121</value>
   <value>0.100815</value>
</cutoff>
```

The third parameter block, which is required to be defined for a full specification of the problem, is the `lattice` node (line 13 in Lst. 1) which describes the lattice graph of the quantum spin model. This parameter block does not actually contain an explicit definition of the lattice (in terms of primitive lattice vectors and basis site positions). Instead, it is assigning values to the `name` attribute and the `range` attribute. The former attribute is a reference to the explicit lattice definition, which is defined in a separate file; for details on the definition of custom lattice graphs, see Sec. 5.2. The latter attribute specifies the truncation range $L$ of vertex functions on the (a priori) infinite lattice graph according to the truncation algorithm described in Sec. 4.1. For example, the parameter block

```
<lattice name="square" range="4"/>
```

would instruct SpinParser to locate and use a lattice implementation with the name "square" and initialize vertex functions to capture two-particle interactions on lattice sites which are up to a maximum of 4 lattice bonds apart.

Lastly, the spin interactions themselves need to be specified, i.e. the values of the interactions constants $J_{ij}^{\mu\nu}$ which appear in the general Hamiltonian Eq. (1) need to be assigned. Similar to the specification of the underlying lattice graph, the structure of the spin interactions is not explicitly defined in the task file. Instead, just like for the lattice, the `model` node must contain an attribute `name`, which references a spin model implementation that is located in a separate file, see Sec. 5.3 for a discussion of custom spin model definitions. Unlike the lattice definition, however, the spin model definition is not exclusively interfaced by the `name` attribute. Rather, spin model definitions may define further custom variables for the interaction constants, which need to be assigned values in the task file. For example, the external spin model definition may implement a nearest-neighbor Heisenberg model with the name "square-heisenberg" and the coupling constant "j". We would instruct the SpinParser software to use this spin model implementation and assign the value 1.0 to the exchange constant "j" with the following parameter block:

```
<model name="square-heisenberg" symmetry="SU2">
    <j>1.0</j>
</model>
```

Note that depending on the specific spin model definition, it is possible that multiple exchange constants are required to be defined, in which case multiple sub-nodes, e.g. "j1" and "j2" for nearest and next-nearest neighbor interactions, respectively, can be appended to the `model` node. Furthermore, the `symmetry` attribute is used to instruct SpinParser on which numerical backend to use for the computation. The SpinParser software provides three different numerical backends which are optimized for different types of spin models: Possible values are (i) "SU2", which supports Heisenberg models (i.e., exchange constants with only $J_{ij}^{xx} = J_{ij}^{yy} = J_{ij}^{zz}$ nonzero), (ii) "XYZ", which supports Kitaev-like models (only $J_{ij}^{xx}$, $J_{ij}^{yy}$, and $J_{ij}^{zz}$ nonzero, but not necessarily all equal), and (iii) "TRI", which supports general time-reversal invariant models as given by the general Hamiltonian Eq. (1). It is possible to facilitate numerical cores with fewer symmetry requirements, e.g. "TRI", for the computation of spin models with greater symmetry, e.g. Heisenberg models, although this would unnecessarily increase the computational cost of the problem. The converse is not true; it is not possible to solve low-symmetry models with a high-symmetry numerical core.

Besides the `parameters` block, which fully specifies the quantum spin model and its numerical solution, it is usually necessary to also define the `measurements` block. The latter contains instructions to extract physical observables from the numerical solution of the quantum spin model. In most cases it is sufficient to specify the spin correlation measurement as

```
<measurement name="correlation"/>
```

which would extract the two-spin correlation function $\chi_{ij}^{\mu\nu,\Lambda}$ as defined in Eq. (29) for all values of $\Lambda$ encountered throughout the evolution of the flow equations (i.e., the values specified in the `cutoff` block of the task file) and for all symmetry-allowed components (i.e., $\chi_{ij}^{xx,\Lambda} = \chi_{ij}^{yy,\Lambda} = \chi_{ij}^{zz,\Lambda}$ for the "SU2" numerical core; $\chi_{ij}^{xx,\Lambda}$, $\chi_{ij}^{yy,\Lambda}$, $\chi_{ij}^{zz,\Lambda}$ for the "XYZ" numerical core; and general $\chi_{ij}^{\mu\nu,\Lambda}$ for $\mu = x, y, z$ if the "TRI" numerical core is selected. All numerical cores further measure the density-like correlations $\chi_{ij}^{00,\Lambda}$.)

A few adjustments are possible in order to further refine the specification of measurements to be recorded. To this end, the `measurement` node can be decorated with additional attributes:

```
<measurement name="correlation" output="measurement.obs" ←
    minCutoff="0.1" maxCutoff="1.0" method="defer"/>
```

The attribute `output` is used to specify the output file where the measurement results are to be stored. Its default output path equals the path of the task file, with the file extension replaced by ".obs". More fine grained control over the cutoff values at which measurements are to be taken is achieved with the `minCutoff` and `maxCutoff` parameters. The former defines a lower limit for the cutoff parameter $\Lambda$ below which no more measurements are taken, whereas the latter defines an upper limit for the cutoff parameter. A special role is assumed by the `method` attribute. Setting this attribute to "defer" instructs the SpinParser software to suppress all measurements and instead write the raw vertex functions to disk. The output file, which contains the vertex data, is written in an HDF5 structure [61], and the file name is generated by replacing the extension of the task file with ".data". The measurements can then be performed later by re-running the SpinParser software with the same task file.

Finally, the task file may contain one more block of information, which is not shown in the example task file Lst. 1, since it is dynamically generated whenever the task file is run in SpinParser; it is the node `calculation`, which contains information on the execution status of the task file. For a new calculation it is simply not present. For a completed calculation, it may look as follows:

```
<calculation startTime="2021-Jan-01 12:00:00" checkpointTime=↩
    "2021-Jan-01 12:10:00" endTime="2021-Jan-01 12:10:00" ↩
    status="finished"/>
```

The attributes `startTime` and `endTime` record the time points at which the computation was started and at which it finished. For computations which may require a long time to solve, the attribute `checkpointTime` is of particular interest. It records the most recent time point at which a so-called checkpoint, i.e. a full snapshot of the current state of the calculation, was written to disk. Such snapshots are periodically written to a file, whose name matches the task file path with the file extension replaced by ".checkpoint", and they allow the resumption of a computation from the state at which the checkpoint was written, even if the SpinParser software was terminated unexpectedly at a later time (e.g. because it exceeded a given computing time allocation.) A computation which was terminated unexpectedly may end up in the calculation status

```
<calculation startTime="2021-Jan-01 12:00:00" checkpointTime=↩
    "2021-Jan-01 12:05:00" status="running"/>
```

Resuming a calculation from a checkpoint is done by simply re-running the SpinParser software with the same task file, which now contains the `status` attribute set to "running". The checkpointing mechanism can also be used in order to extend a previously completed calculation down to lower cutoff values of the cutoff parameter: Simply edit the task file to contain a lower minimum cutoff value in the `parameters` section and manually set the `status` to "running".

## 5.2 Definition of lattice graphs

The lattice graph is an integral part of the definition of a quantum spin model in the form of the general Hamiltonian Eq. (1). On one and the same lattice graph, it is possible to define a plethora of different spin models with different spin interactions, so in order to reduce redundancies in the implementation of quantum spin models, it is reasonable to separate the lattice definition from the definition of spin interactions. In this subsection, we outline the implementation of lattices in the SpinParser software.

We mentioned previously, in Sec. 5.1, that lattice definitions are only referenced in the task file, whereas the actual definition happens elsewhere. In fact, lattice definitions are read from external files with XML-like structure, see the full example in Lst. 2. We refer to those files as "resource files". Resource files are automatically searched for lattice definitions whose name matches the one specified in the task file. Any files with the file ending ".xml" within the search path are automatically searched; the search path for resource files can be provided as a command

```
1  <unitcell name="square">
2     <primitive x="1" y="0" z="0" />
3     <primitive x="0" y="1" z="0" />
4     <primitive x="0" y="0" z="1" />
5
6     <site x="0" y="0" z="0" />
7
8     <bond from="0" to="0" da0="1" da1="0" da2="0" />
9     <bond from="0" to="0" da0="0" da1="1" da2="0" />
10 </unitcell>
11
12 <unitcell name="honeycomb">
13    <primitive x="3/2" y="sqrt(3)/2" z="0" />
14    <primitive x="3/2" y="-sqrt(3)/2" z="0" />
15    <primitive x="0" y="0" z="1" />
16
17    <site x="0" y="0" z="0" />
18    <site x="1" y="0" z="0" />
19
20    <bond from="0" to="1" da0="0" da1="0" da2="0" />
21    <bond from="1" to="0" da0="1" da1="0" da2="0" />
22    <bond from="1" to="0" da0="0" da1="1" da2="0" />
23 </unitcell>
```

Listing 2: **Examples of lattice definitions**. Every lattice is constructed from a unit cell, which is repeated periodically according to the shift defined by the primitive lattice vectors. Multiple unit cell definitions can be summarized in one file, as illustrated here for the square lattice (lines 1–10) and the honeycomb lattice (lines 12–23).

line option to the SpinParser executable, see Sec. 5. If no search path is specified explicitly, it defaults to the first existing directory from the following two:

```
$BINDIR/../res
$BINDIR/
```

where $BINDIR is the directory in which the SpinParser executable is located.

The definition of every lattice is based on the full specification of a single unit cell (cf. lines 12–23 in Lst. 2). One unit cell hereby refers to the set of primitive lattice vectors which define the periodicity of the underlying Bravais lattice, any number of basis sites, and all lattice bonds associated with that unit cell. The unit cell definition itself

```
<unitcell name="honeycomb">
   <!-- definition of primitives, basis sites, and lattice ←
      bonds goes here -->
</unitcell>
```

hereby includes an attribute name ("honeycomb", in this case), which is referenced in the task file and used to identify the matching lattice definition. The three primitive Bravais lattice vectors

```
<primitive x="3/2" y="sqrt(3)/2" z="0" />
<primitive x="3/2" y="-sqrt(3)/2" z="0" />
<primitive x="0" y="0" z="1" />
```

are defined as Cartesian three-dimensional vectors via their $x$, $y$, and $z$ components, which are listed as attributes of the respective `primitive` nodes. The lattice primitives have an implicit order in which they are defined; we shall refer to them as the zeroth, the first, and the second lattice vector, respectively, which in the example above are

$$
a_0 = \begin{pmatrix} 3/2 \\ \sqrt{3}/2 \\ 0 \end{pmatrix}, \quad a_1 = \begin{pmatrix} 3/2 \\ -\sqrt{3}/2 \\ 0 \end{pmatrix}, \quad a_2 = \begin{pmatrix} 0 \\ 0 \\ 1 \end{pmatrix}. \tag{45}
$$

Note that the definition of every lattice unit cell is embedded into a three-dimensional space, regardless of the dimensionality of the lattice itself. Two-dimensional lattices simply do not implement any lattice bonds along the third dimension (see below).

Basis sites are defined by `site` nodes, which comprise $x$, $y$, and $z$ components that describe the position of the basis site within the unit cell:

```
<site x="0" y="0" z="0" />
<site x="1" y="0" z="0" />
```

Every unit cell definition may contain any number of basis sites. Basis sites are implicitly indexed according to the order in which they are defined, starting at zero. In the example above, the two basis sites

$$
b_0 = \begin{pmatrix} 0 \\ 0 \\ 0 \end{pmatrix}, \quad b_1 = \begin{pmatrix} 1 \\ 0 \\ 0 \end{pmatrix} \tag{46}
$$

are defined, forming the two-site basis of the bipartite honeycomb lattice.

Finally, the connectivity of the lattice needs to be established. To this end, a list of all the lattice bonds within a single unit cell must be provided:

```
<bond from="0" to="1" da0="0" da1="0" da2="0" />
<bond from="1" to="0" da0="1" da1="0" da2="0" />
<bond from="1" to="0" da0="0" da1="1" da2="0" />
```

The definition of a lattice bond is to be understood as follows. Every lattice bond connects two lattice sites, which are specified via the `from` and the `to` attribute. The values of these attributes refer to the index of the basis site, i.e., the first bond in the example above would connect basis site $b_0$ to basis site $b_1$. This first bond connects two sites within the same unit cell; however, we also need to specify the connections to neighboring unit cells. For this purpose the attributes `da0`, `da1`, and `da2` exist, which capture the offset of the target lattice site in units of the primitive lattice vectors $a_0$, $a_1$, and $a_2$, respectively. The second lattice bond in the example above would therefore connect basis site $b_1$ of one unit cell to basis site $b_0$ of the unit cell shifted by the lattice vector $a_0$. Similarly, the third lattice bond connects basis site $b_1$ of one unit cell to basis site $b_0$ of the unit cell shifted by the lattice vector $a_1$. In deciding which lattice bonds to include in the unit cell definition, one needs to take care not to double count bonds. Each bond, which connects sites in between two unit cells, must only be attributed to one of the two neighboring unit cells, see the illustration in Fig. 7.

## 5.3 Definition of spin interactions

With the definition of the underlying lattice graph discussed in the previous section, the second integral part to defining a quantum spin model is the specification of the interaction constants $J_{ij}^{\mu\nu}$ themselves. These are defined in close analogy to the lattice unit cells; they are defined in XML-like resource files located within the same search path as for the lattice unit cell definitions (cf. Sec. 5.2. Note, however, that while lattice unit cells are self-sustained objects, the definition of a

(a)
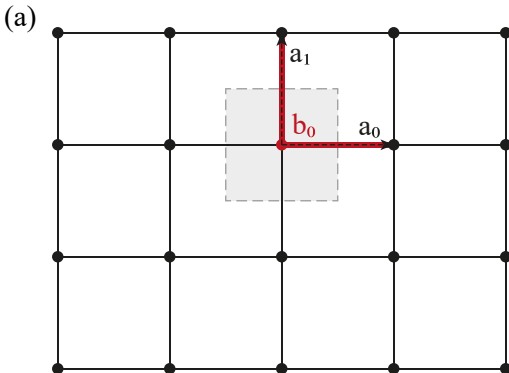

(b)
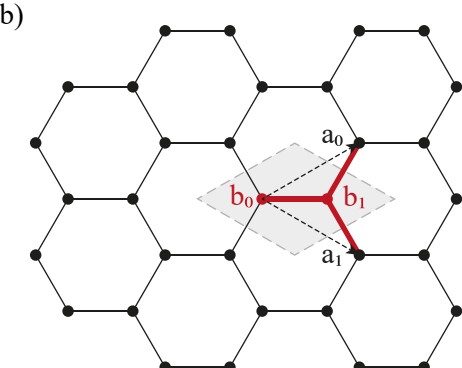

Figure 7: **Lattice unit cell definitions** for the square lattice and the honeycomb lattice. (a) The unit cell of the square lattice (gray shaded square) contains the two primitive lattice vectors $a_0$ and $a_1$, one lattice site (site $b_0$, indicated in red), and two lattice bonds (indicated in red). (b) The unit cell of the honeycomb lattice (gray shaded diamond) contains the the two primitive lattice vectors $a_0$ and $a_1$, two lattice sites (sites $b_0$ and $b_1$, indicated in red), and three lattice bonds (indicated in red).

spin model is less general. It is typically tailored to a specific underlying lattice, since in its definition we make use of the concrete lattice parametrization and the list of basis sites. Full examples of spin model definitions for the Heisenberg model on the square lattice and for the Heisenberg-Kitaev model on the honeycomb lattice are shown in Lst. 3 (lines 1–4 and 6–13, respectively). Every spin model definition is effectively a list of two-spin interaction terms, each represented by an `interaction` node in the XML structure. Each two-spin interaction is fully characterized by the two lattice sites it connects, the information on which spin components are being coupled, and a name for the coupling constant to be referenced in the task file for setting the actual value of the interaction strength.

The two connecting lattice sites are specified via the attributes `from` and `to`. Each of these attributes is to be assigned a tuple of four comma separated values which reference lattice sites by the lattice vectors $a_0$, $a_1$, $a_2$ and the basis site index $b$. For example, the interaction

```
<interaction parameter="j" from="0,0,0,0" to="1,0,0,0" type="↩
    heisenberg" />
```

would couple the lattice site at $0 \cdot a_0 + 0 \cdot a_1 + 0 \cdot a_2 + b_0$ with the lattice site at $1 \cdot a_0 + 0 \cdot a_1 + 0 \cdot a_2 + b_0$; in the definition of the square lattice shown in Lst. 2 these two lattice sites would be nearest neighbors. All interactions within one unit cell of the lattice spin model need to be defined – similar to the definition of a lattice unit cell, care needs to be taken in the definition in order to avoid double counting of interactions. Furthermore, we emphasize that unlike the lattice bond definitions, spin interaction definitions have a sense of orientation, i.e., for inversion-symmetry breaking interactions (Dzyaloshinskii-Moriya interactions) it sometimes may be convenient to define the first lattice site (the `from` attribute) to lie outside of the reference unit cell and the second site (the `to` attribute) to be within the reference unit cell (whereas in the definition of lattice bonds definitions, the first site always lies within the reference unit cell.)

The information of which spin components should be coupled is contained in the `type` attribute. All spin interactions are of the form

$$\mathbf{S}_{\text{from}} \cdot \mathbf{M} \cdot \mathbf{S}_{\text{to}}, \tag{47}$$

where $\mathbf{S}_{\text{from}}$ and $\mathbf{S}_{\text{to}}$ are the two spins involved, and the $3 \times 3$ matrix $\mathbf{M}$ determines the structure of the interaction. The `type` attribute assumes one of the following possible string values:

```
1  <model name="square-heisenberg">
2     <interaction parameter="j" from="0,0,0,0" to="1,0,0,0" ↵
          type="heisenberg" />
3     <interaction parameter="j" from="0,0,0,0" to="0,1,0,0" ↵
          type="heisenberg" />
4  </model>
5
6  <model name="honeycomb-kitaev">
7     <interaction parameter="j" from="0,0,0,0" to="0,0,0,1" ↵
          type="heisenberg" />
8     <interaction parameter="j" from="0,0,0,0" to="0,-1,0,1" ↵
          type="heisenberg" />
9     <interaction parameter="j" from="0,0,0,0" to="-1,0,0,1" ↵
          type="heisenberg" />
10    <interaction parameter="k" from="0,0,0,0" to="0,0,0,1" ↵
          type="xx" />
11    <interaction parameter="k" from="0,0,0,0" to="0,-1,0,1" ↵
          type="yy" />
12    <interaction parameter="k" from="0,0,0,0" to="-1,0,0,1" ↵
          type="zz" />
13 </model>
```

Listing 3: **Example of spin model definitions**. Every spin model contains a list of two-spin interactions, each of which is characterized by the two lattice sites it connects as well as the spin components it couples. Multiple spin model definitions can be summarized in one file, as illustrated here for the Heisenberg model on the square lattice (lines 1–4) and the Heisenberg-Kitaev model on the honeycomb lattice (lines 6–13).

"heisenberg", "xxyy", "gx", "gy", "gz", "$\mu\nu$" or "$-\mu\nu$", where $\mu$ and $\nu$ are either $x$, $y$ or $z$. The first two string values resemble Heisenberg and XY interactions, respectively, and translate into the interaction matrices

$$\mathbf{M}_{\mathrm{heisenberg}} = \begin{pmatrix} 1 & 0 & 0 \\ 0 & 1 & 0 \\ 0 & 0 & 1 \end{pmatrix} \quad \text{and} \quad \mathbf{M}_{\mathrm{xxyy}} = \begin{pmatrix} 1 & 0 & 0 \\ 0 & 1 & 0 \\ 0 & 0 & 0 \end{pmatrix}. \tag{48}$$

Symmetric off-diagonal interactions, typically referred to as $\Gamma$-interactions in the literature, are accessible via the string values "gx", "gy", and "gz", and translate into the respective interaction matrices

$$\mathbf{M}_{\mathrm{gx}} = \begin{pmatrix} 0 & 0 & 0 \\ 0 & 0 & 1 \\ 0 & 1 & 0 \end{pmatrix}, \quad \mathbf{M}_{\mathrm{gy}} = \begin{pmatrix} 0 & 0 & 1 \\ 0 & 0 & 0 \\ 1 & 0 & 0 \end{pmatrix}, \quad \text{and} \quad \mathbf{M}_{\mathrm{gz}} = \begin{pmatrix} 0 & 1 & 0 \\ 1 & 0 & 0 \\ 0 & 0 & 0 \end{pmatrix}. \tag{49}$$

Finally, the string values "$\mu\nu$" and "$-\mu\nu$" denote specific two-spin interactions between the $\mu$ and $\nu$ components of spins $\mathbf{S}_{\mathrm{from}}$ and $\mathbf{S}_{\mathrm{to}}$; their matrix components are given by

$$(\mathbf{M}_{\mu\nu})_{\alpha\beta} = \delta_{\alpha\beta} \quad \text{and} \quad (\mathbf{M}_{-\mu\nu})_{\alpha\beta} = -\delta_{\alpha\beta}. \tag{50}$$

Since it is possible to define multiple types of interactions between the same two lattice sites, the latter two expressions can be used as building blocks to construct more complicated exchange terms, e.g. Dzyaloshinskii-Moriya or $\Gamma'$ interactions, which sometimes emerge in the simulation of Hamiltonians for realistic materials.

## 5.4  Output data

In the previous subsections, we have discussed in detail how quantum spin models can be set up for a numerical solution with the help of SpinParser. In this section, we discuss the next crucial step: the interpretation of the output data. Besides the temporary ".data" and ".checkpoint" files mentioned before, running the SpinParser software produces two more key output files – the actual measurement output (the ".obs" file mentioned in Sec. 5.1) and a second file, which contains a description of the lattice spin model which was constructed based on the parameter specifications in the task file and the resource files. The latter is stored in a file whose name is generated by substituting the file name extension of the task file with the ending ".ldf".

The lattice description file ".ldf" is simply an XML-structured list of all lattice sites, lattice bonds, and spin interactions within the lattice with additional information on the symmetry reduction which has been performed by SpinParser. The purpose of the lattice description file is to enable a simple assessment of whether the lattice spin model and the coupling constants specified in the task file and in the resource files have been implemented correctly. To this end, it is helpful to import the ".ldf" file in an automated script for further processing and subsequent plotting. Note that while the SpinParser executable itself does not offer further processing or visualization of ".ldf" files, the SpinParser source code distribution [62] contains additional Python scripts for the visualization of ".ldf" files. A full example of a lattice description file is shown in Lst. 4.

Lattice sites are specified in the ".ldf" file as XML nodes of the name `site`, and each lattice site is assigned a unique identifier via the attribute `id`:

```
<site id="0" x="0.000000" y="0.000000" z="0.000000" ↩
    parametrized="true"/>
```

Furthermore, the real space position (embedded in Cartesian three-dimensional space) of each lattice site is stored in the three attributes `x`, `y`, and `z`, which allows to easily generate a visual representation of the lattice. The last attribute, `parametrized`, carries information on the internal representation of the lattice spin model, which the SpinParser software has generated in order to optimize the computation: If the value is "true", the lattice site is part of the internal computational basis and all vertex functions are computed explicitly for that lattice site. If the value is "false", vertex functions which involve that lattice site are not evaluated explicitly, but rather they are related to the reduced internal computational basis via lattice and/or spin symmetries. The ratio of the total number of lattice sites $N_s^{\text{total}}$ (parametrized and unparametrized) to the number of parametrized lattice sites $N_s$ thus serves as an indicator of the numerical complexity of the problem and the simplification which was achieved by exploiting symmetry transformations (cf. the discussion in Sec. 4.1.) Lattice bonds

```
<bond from="0" to="13" />
```

are specified in reference to the unique identifiers of the connecting lattice sites, which are stored in the `from` and `to` attributes. Lattice bonds reflect the connectivity of the lattice sites and are independent from the spin interactions in the model. The latter are defined by `interaction` nodes in the XML structure,

```
<interaction from="0" to="13" value="↩
    [[1.000000,0.000000,0.000000], ↩
    [0.000000,1.000000,0.000000], ↩
    [0.000000,0.000000,1.000000]]" />
```

which, similar to the lattice bonds, reference two lattice sites between which the interactions occur. The interaction type and strength is encoded in a $3 \times 3$ interaction matrix, akin to the definition in Eq. (47), whose entries are stored in the `value` attribute. The values are stored row-wise, i.e., the three tuples of values resemble the three rows of the interaction matrix.

```
1  <lattice>
2      <site id="0" x="0.000000" y="0.000000" z="0.000000" ↵
           parametrized="true"/>
3      <site id="1" x="0.000000" y="-1.000000" z="0.000000" ↵
           parametrized="true"/>
4      <site id="2" x="0.000000" y="-2.000000" z="0.000000" ↵
           parametrized="true"/>
5      <!-- [...] more sites may be listed here -->
6      <bond from="0" to="13" />
7      <bond from="0" to="14" />
8      <bond from="1" to="0" />
9      <!-- [...] more bonds may be listed here -->
10     <interaction from="0" to="13" value="↵
           [[1.000000,0.000000,0.000000], ↵
           [0.000000,1.000000,0.000000], ↵
           [0.000000,0.000000,1.000000]]" />
11     <interaction from="0" to="14" value="↵
           [[1.000000,0.000000,0.000000], ↵
           [0.000000,1.000000,0.000000], ↵
           [0.000000,0.000000,1.000000]]" />
12     <interaction from="1" to="0" value="↵
           [[1.000000,0.000000,0.000000], ↵
           [0.000000,1.000000,0.000000], ↵
           [0.000000,0.000000,1.000000]]" />
13     <!-- [...] more interactions may be listed here -->
14  </lattice>
```

Listing 4: **Example of a lattice description file**. The file has a simple XML structure, in which all lattice sites within truncation range of a fixed reference site are listed (lines 2–5), all bonds connecting the lattice sites are specified (lines 6–9), and all spin interactions are detailed with their interaction strength (lines 10–13).

The second (and arguably more important) output file is the ".obs" file, in which the observed measurement results are stored. The precise content of the ".obs" file depends on the details of the task file – in particular on the `measurements` block in the task file, see Sec. 5.1, and on the selected numerical core in the `model` section of the task file. However, as we shall see below, the general structure of the correlation measurement output is always the same, regardless of the choice of the numerical core. The measurement data is stored in an HDF5 format. An exemplary file structure, which was generated for a Heisenberg model on the square lattice with the numerical backend for SU(2) symmetric models, is shown in Lst. 5. We emphasize the use of the numerical core for SU(2)-symmetric models here, because it directly affects the output file: Depending on the symmetry of the model, different components of the two-spin correlation function are measured, see the paragraph on measurements in Sec. 5.1. For the "SU2" numerical core, the root path in the HDF5 file structure contains the two groups

```
/SU2CorDD                                          Group
/SU2CorZZ                                          Group
```

which correspond to the two-spin correlation measurements $\chi_{ij}^{00,\Lambda}$ and $\chi_{ij}^{zz,\Lambda}$, respectively. Analogously, for the "XYZ" numerical core, the groups `XYZCorDD`, `XYZCorXX`, `XYZCorYY`, and `XYZCorZZ` would be created. In case of the most general "TRI" numerical core, the groups

```
 1  /                                              Group
 2  /SU2CorDD                                      Group
 3  /SU2CorDD/data                                 Group
 4  /SU2CorDD/data/measurement_0                   Group
 5  /SU2CorDD/data/measurement_0/cutoff            Attribute {1}
 6  /SU2CorDD/data/measurement_0/data              Dataset {1, 41}
 7  /SU2CorDD/data/measurement_1                   Group
 8  /SU2CorDD/data/measurement_1/cutoff            Attribute {1}
 9  /SU2CorDD/data/measurement_1/data              Dataset {1, 41}
10  # [...] more measurements may be listed here
11  /SU2CorDD/meta                                 Group
12  /SU2CorDD/meta/basis                           Dataset {1}
13  /SU2CorDD/meta/latticeVectors                  Dataset {3}
14  /SU2CorDD/meta/sites                           Dataset {1, 41}
15  /SU2CorZZ                                      Group
16  /SU2CorZZ/data                                 Group
17  /SU2CorZZ/data/measurement_0                   Group
18  /SU2CorZZ/data/measurement_0/cutoff            Attribute {1}
19  /SU2CorZZ/data/measurement_0/data              Dataset {1, 41}
20  /SU2CorZZ/data/measurement_1                   Group
21  /SU2CorZZ/data/measurement_1/cutoff            Attribute {1}
22  /SU2CorZZ/data/measurement_1/data              Dataset {1, 41}
23  # [...] more measurements may be listed here
24  /SU2CorZZ/meta                                 Group
25  /SU2CorZZ/meta/basis                           Dataset {1}
26  /SU2CorZZ/meta/latticeVectors                  Dataset {3}
27  /SU2CorZZ/meta/sites                           Dataset {1, 41}
```

Listing 5: **Example structure of an observable output file**. The data shown here is the output which is generated from the task file shown in Lst. 1. The output file is stored in HDF5 format. The two groups `SU2CorDD` and `SU2CorZZ` (lines 2 and 15) contain the correlation measurements $\chi_{ij}^{00,\Lambda}$ and $\chi_{ij}^{zz,\Lambda}$, respectively. Each measurement is composed of meta information about the lattice (lines 11–14 and 24–27), as well as the measurement data itself (lines 3–10 and 16–23). See text for details.

`TRICorDD` and `TRICorAB` for any A,B=X,Y,Z would be created.

Each such group of measurements contains additional meta information on the underlying lattice geometry, e.g.

```
/SU2CorDD/meta/basis                           Dataset {1}
/SU2CorDD/meta/latticeVectors                  Dataset {3}
/SU2CorDD/meta/sites                           Dataset {1, 41}
```

where the dataset `basis` in this case is a one-element list of a three-component vector, which contains the real-space position of the first (and only) basis site in the unit cell of the square lattice. For lattices with $N_b$ basis sites, the dataset would be a list of $N_b$ three-component vectors capturing the positions of the basis sites $b_0, \ldots, b_{N_b-1}$. The dataset `latticeVectors` is a three-element list of three-component vectors, which contains the lattice vectors $a_0$, $a_1$, and $a_2$, respectively. The third dataset in the meta information, `sites`, is an $N_b \times N_s^{\text{total}}$ matrix of three-component vectors, where $N_s^{\text{total}}$ is the number of lattice sites within the truncation range around a given reference site (cf. Sec. 4.1). The matrix contains, for every basis site in the lattice, the list

of real-space positions of all lattice sites within the truncation range around that basis site.

The same order of lattice sites, as stored in the $N_b \times N_s^{\text{total}}$ matrices described above, is being used for the storage of the actual correlation measurement results – the purpose of the meta data is to serve as a label for the measurement data. Any two-spin correlations, e.g. $\chi_{ij}^{zz,\Lambda}$, are measured for lattice site $i$ running over all basis sites of the lattice and lattice site $j$ running over all sites within the truncation range around site $i$, thus resulting in an $N_b \times N_s^{\text{total}}$ matrix of correlation data. Every measurement is stored in an HDF5 group of the form

```
/SU2CorZZ/data/measurement_0              Group
/SU2CorZZ/data/measurement_0/cutoff       Attribute {1}
/SU2CorZZ/data/measurement_0/data         Dataset {1, 41}
```

where the integer number trailing the group name `measurement_0` is simply a unique identifier number. The correlation data is stored in the $N_b \times N_s^{\text{total}}$ dataset `data`, while the cutoff value of the correlation measurement is stored in the attribute `cutoff`. Throughout the solution of the pf-FRG flow equations, whenever a measurement is recorded at some cutoff value $\Lambda$, a new data group with a new unique identifier is created in the ".obs" file.

# 6   Examples

In the previous sections we have discussed the underlying theory of the pseudofermion functional renormalization group, its numerical implementation, and the general usage of the SpinParser software. We round up the presentation in this section by showing two fully worked out examples on how to study aspects of quantum magnetism with the help of the SpinParser software. The first example, which we discuss in Sec. 6.1, illustrates the use of the SpinParser software for the analysis of a three-dimensional quantum spin model of competing nearest and next-nearest neighbor Heisenberg interactions on the cubic lattice. Depending on the ratio of the two types of interactions, the model can host different magnetization textures in its ground state. In the second example, Sec. 6.2, we showcase a model with more intricate spin interactions which break the inversion symmetry of lattice bonds, namely the kagome Heisenberg antiferromagnet with additional Dzyaloshinskii-Moriya (DM) interactions. Such competing interactions can naturally arise in realistic materials which attempt to emulate the kagome Heisenberg antiferromagnet, and their presence raises the question of how stable the spin liquid ground state of the pure kagome Heisenberg antiferromagnet is against perturbation by DM interactions.

## 6.1   J1-J2-Heisenberg model on the cubic lattice

One of the simplest spin model Hamiltonians to suppress magnetic order in the ground state is the $J_1$-$J_2$-Heisenberg model on the two-dimensional square lattice. The suppression of magnetic long range order on the bipartite square lattice is due to the competition of antiferromagnetic nearest neighbor and next-nearest neighbor interactions, which are inherently incompatible with each other. While the nearest neighbor interactions would favor a Néel ground state, the next-nearest neighbor interactions would favor collinear antiferromagnetic order. As a consequence, if neither one of the interactions is clearly dominant, and the ratio of nearest neighbor interactions $J_1$ and next-nearest neighbor interactions $J_2$ is approximately $J_2/J_1 = 0.5$, the system is found to exhibit a paramagnetic ground state (with details about the precise nature of that ground state phase diagram still under debate) [5, 18].

In this example, we consider the extension of the aforementioned $J_1$-$J_2$-Heisenberg model onto the three-dimensional simple cubic lattice. Our model is captured by the microscopic Hamil-

(a)
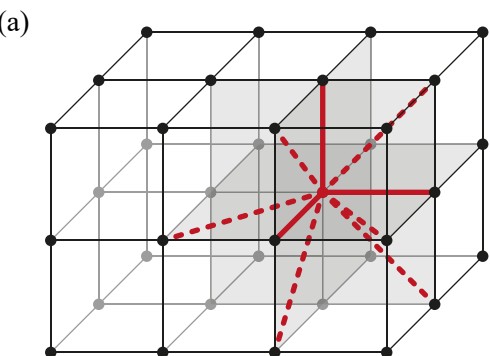

(b)
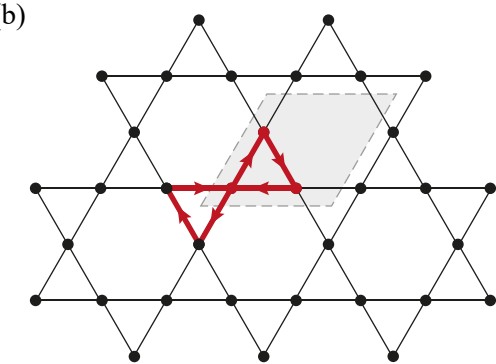

Figure 8: **Spin model definitions** on the cubic lattice and on the kagome lattice. (a) The $J_1$-$J_2$-Heisenberg model on the cubic lattice is defined by a single-site lattice unit cell (site indicated in red), three nearest neighbor interactions per unit cell (thick red lines), and six next-nearest neighbor interactions per unit cell (dashed red lines). In total, each site has 6 nearest neighbors and 12 next-nearest neighbors (4 in each gray-shaded plane). (b) The Heisenberg-DM model on the kagome lattice has a three-site unit cell (red sites) with six nearest-neighbor interactions per unit cell (thick red lines). The DM interactions $\widehat{\mathbf{D}}_{ij} \cdot (\mathbf{S}_i \times \mathbf{S}_j)$ are directed from site $i$ to $j$ according to the red arrows, with the DM vector $\widehat{\mathbf{D}}$ pointing out of the plane for all interactions.

tonian

$$H = J_1 \sum_{\langle i,j \rangle} \mathbf{S}_i \mathbf{S}_j + J_2 \sum_{\langle\langle i,j \rangle\rangle} \mathbf{S}_i \mathbf{S}_j \,, \tag{51}$$

where the first sum runs over all pairs of nearest neighbor sites in the cubic lattice and the second sum runs over pairs of next-nearest neighbors, see Fig. 8a. With the increased spatial dimension leading to a greater number of interacting neighbor sites – 6 nearest neighbors and 12 next-nearest neighbors, compared to 4 nearest and next-nearest neighbors each on the square lattice – one expects a stronger tendency of the system towards the formation of magnetic order. Indeed, it is observed both in the classical [63] and quantum [29,64] version of the model that there is a direct transition from the Néel ground state configuration at dominant $J_1$ into the collinear antiferromagnetic order at large $J_2$.

Our goal is to confirm the existence of the transition from the Néel state to the collinear antiferromagnetic order with the help of the SpinParser software. The model Hamiltonian Eq. (51) has previously been investigated in Ref. [29] by means of an independent implementation of the pf-FRG algorithm, which allows us to benchmark our results. In order to set up the calculation, we define the cubic lattice as well as the $J_1$-$J_2$-Heisenberg spin model in the XML structures described in Secs. 5.2 and 5.3, respectively:

```
<unitcell name="cubic">
    <primitive x="1" y="0" z="0" />
    <primitive x="0" y="1" z="0" />
    <primitive x="0" y="0" z="1" />

    <site x="0" y="0" z="0" />

    <bond from="0" to="0" da0="1" da1="0" da2="0" />
    <bond from="0" to="0" da0="0" da1="1" da2="0" />
    <bond from="0" to="0" da0="0" da1="0" da2="1" />
</unitcell>
```

```
<model name="cubic-j1j2heisenberg">
   <interaction parameter="j1" from="0,0,0,0" to="1,0,0,0" ↩
      type="heisenberg" />
   <interaction parameter="j1" from="0,0,0,0" to="0,1,0,0" ↩
      type="heisenberg" />
   <interaction parameter="j1" from="0,0,0,0" to="0,0,1,0" ↩
      type="heisenberg" />
   <interaction parameter="j2" from="0,0,0,0" to="0,1,-1,0" ↩
      type="heisenberg" />
   <interaction parameter="j2" from="0,0,0,0" to="0,1,1,0" ↩
      type="heisenberg" />
   <interaction parameter="j2" from="0,0,0,0" to="1,-1,0,0" ↩
      type="heisenberg" />
   <interaction parameter="j2" from="0,0,0,0" to="1,0,-1,0" ↩
      type="heisenberg" />
   <interaction parameter="j2" from="0,0,0,0" to="1,0,1,0" ↩
      type="heisenberg" />
   <interaction parameter="j2" from="0,0,0,0" to="1,1,0,0" ↩
      type="heisenberg" />
</model>
```

In fact, the definition of both the lattice and the spin model are already included in the default installation of the SpinParser software; they are located in the files `$SPINPARSER/res/lattices.xml` and `$SPINPARSER/res/models.xml`, respectively, where `$SPINPARSER` is the root directory of the SpinParser installation.

In the next step, we set up the actual task file for the calculation as detailed in Sec. 5.1. To this end, we prepare the file `$SPINPARSER/examples/cubic-j1j2.xml` with the following content:

```
<task>
   <parameters>
      <frequency discretization="exponential">
         <min>0.005</min>
         <max>50.0</max>
         <count>64</count>
      </frequency>
      <cutoff discretization="exponential">
         <max>50.0</max>
         <min>0.01</min>
         <step>0.98</step>
      </cutoff>
      <lattice name="cubic" range="7"/>
      <model name="cubic-j1j2heisenberg" symmetry="SU2">
         <j1>1.000000</j1>
         <j2>0.000000</j2>
      </model>
   </parameters>
   <measurements>
      <measurement name="correlation"/>
   </measurements>
</task>
```

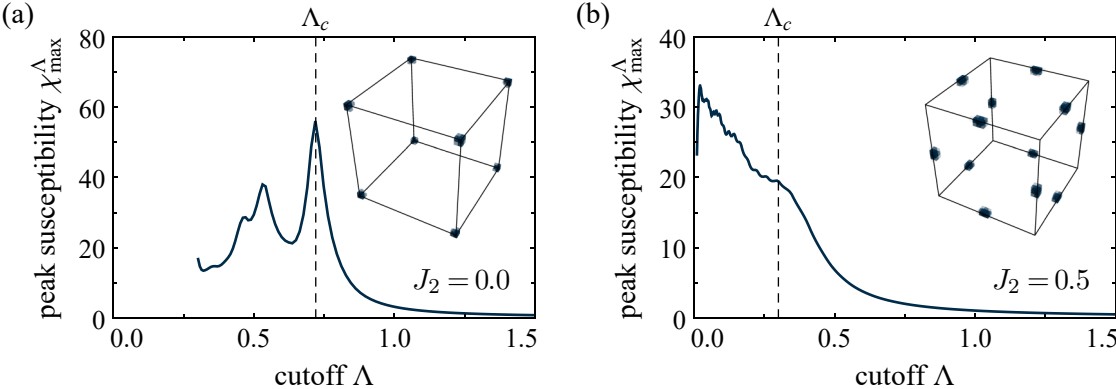

Figure 9: **Renormalization group flow** of the two-spin correlations in the $J_1$-$J_2$-Heisenberg model on the cubic lattice. (a) Flow of the peak susceptibility at fixed $J_1 = 1.0$ and vanishing next-nearest neighbor interaction $J_2 = 0$, exhibiting a flow breakdown at $\Lambda_c = 0.72$. (b) Flow of the peak susceptibility at $J_2 = 0.5$, with a breakdown at $\Lambda_c = 0.3$. The insets indicate regions within the Brillouin zone where the intensity of the structure factor is within the top 20% of its maximum value, plotted at $\Lambda_c$ for (a) $J_2 = 0.0$ and (b) $J_2 = 0.5$.

The parameters chosen here, i.e., a frequency discretization of $N_\omega^{\text{total}} = 128$ points ($N_\omega = 64$ positive frequencies) and a lattice truncation range of $L = 7$ bonds, are typically expected to give good results. Yet, in order to speed up the calculation for phase diagram scans, it might be appropriate to perform calculations at reduced precision; similarly, in order to check the convergence for production-quality calculations, it might be advised to increase the lattice truncation range or the frequency discretization. We defined exchange constants $J_1 = 1.0$ and $J_2 = 0.0$ in the example task file above – these values are of course only exemplary and should be adjusted as needed. We are now prepared run the actual calculation by invoking the command

```
$SPINPARSER/bin/SpinParser $SPINPARSER/examples/cubic-j1j2.↩
   xml
```

which produces the HDF5-structured output file at `$SPINPARSER/examples/cubic-j1j2.obs`. Executing the calculation takes approximately 25 core hours on an Intel Xeon Phi 7250-F Knights Landing processor (68 physical cores, 1.4 GHz).

We extract the two-spin correlation function $\chi_{ij}^{zz,\Lambda}$ from the HDF5 datasets `/SU2CorZZ/data/measurement_*/data` within the result file (see Sec. 5.4 for details), and with the help of the lattice site positions stored in the dataset `/SU2CorZZ/meta/sites` we perform a Fourier transformation to obtain the momentum-resolved structure factor $\chi^\Lambda(\mathbf{k})$ as defined in Eq. (31). Comparing the results for $J_2 = 0$ and $J_2 = 0.5$, we observe decisively different magnetic ordering patterns: At vanishing next-nearest neighbor interactions, the ground state order is simply the Néel configuration, which is associated with structure factor peaks at the corners of the Brillouin zone (see inset of Fig. 9a). Once $J_2$ becomes sizable, at $J_2 = 0.5$, the ground state is formed by collinear order, which consists of an antiferromagnetic arrangement of ferromagnetic columns of spins. Its structure factor exhibits peaks on the centers of the Brillouin zone edges (inset of Fig. 9b).

A key indicator for the nature of the ground state of the system in pf-FRG calculations is the qualitative behavior of the flow, discriminating whether it runs smoothly to the lowest cutoff value or whether it exhibits a breakdown of the smooth flow at some finite critical cutoff $\Lambda_c$. The latter scenario is associated with spontaneous symmetry breaking and the onset of magnetic order, which we expect to observe on the cubic lattice. Indeed, when plotting the evolution of the dominant component in the structure factor as a function of the cutoff $\Lambda$, a pronounced breakdown becomes visible at $\Lambda_c = 0.72$ for $J_2 = 0.0$ (Fig. 9a). Similarly, in the collinearly ordered ground

state at $J_2 = 0.5$, a flow breakdown manifests (although less pronounced) at $\Lambda_c = 0.3$. The same values for the critical cutoff $\Lambda_c$ have previously been identified in Ref. [29] in the context of the extended $J_1$-$J_2$-$J_3$-Heisenberg model on the cubic lattice.

## 6.2 Kagome antiferromagnet with Dzyaloshinskii-Moriya interactions

We now consider a second example, which sets itself apart from the previous one by its greatly reduced symmetries. On top of the SU(2)-symmetric Heisenberg interactions, which we have seen in the previous example, we now introduce Dzyaloshinskii-Moriya (DM) interactions, which break the lattice inversion symmetry as well as the SU(2) spin rotation symmetry of the Hamiltonian. Specifically, we consider the kagome Heisenberg antiferromagnet augmented by DM interactions, which can be captured by the Hamiltonian

$$H = J \sum_{\langle i,j \rangle} \mathbf{S}_i \mathbf{S}_j + D \sum_{\langle i,j \rangle} \widehat{\mathbf{D}}_{ij} \cdot (\mathbf{S}_i \times \mathbf{S}_j) , \tag{52}$$

where we fix $J = 1$ and the sums run over all pairs of nearest neighbor sites $i$ and $j$ on the kagome lattice with a sense of direction as indicated in Fig. 8b, with the DM vectors $\widehat{\mathbf{D}}_{ij}$ pointing out of the lattice plane and having unit length.

It is well agreed upon that the kagome Heisenberg antiferromagnet, which is the limiting case of our model Hamiltonian Eq. (52) at $D = 0$, harbors a quantum spin liquid ground state at low temperatures – although the precise nature of the spin liquid state remains under debate [5]. With the microscopic Hamiltonian of the kagome Heisenberg antiferromagnet being strikingly simple, much effort went into the search for material candidates which could potentially realize the model. One of the hitherto cleanest material realizations is the so-called herbertsmithite compound [65]. However, even herbertsmithite is not a perfect realization of the kagome Heisenberg antiferromagnet, and finite DM interactions beyond the dominant Heisenberg exchange terms can be expected [66,67]. From a theoretical perspective it is thus interesting to study the stability of the spin liquid ground state of the unperturbed kagome Heisenberg antiferromagnet against finite DM interactions [47,48,68,69]. Such analysis has been performed by means of the pf-FRG approach in Refs. [47] and [48]. Here, we review the calculations as an example for a model which breaks both spin rotation symmetry and lattice inversion symmetry – an intricate model, which spotlights the wide applicability of the SpinParser software.

The calculation is performed as follows. First, we define the kagome lattice as well as the Heisenberg-DM spin model in the XML structures described in Secs. 5.2 and 5.3, respectively:

```
<unitcell name="kagome">
   <primitive x="2" y="0" z="0" />
   <primitive x="1" y="sqrt(3)" z="0" />
   <primitive x="0" y="0" z="1" />

   <site x="0" y="0" z="0" />
   <site x="1" y="0" z="0" />
   <site x="1/2" y="sqrt(3)/2" z="0" />

   <bond from="0" to="1" da0="0" da1="0" da2="0" />
   <bond from="0" to="2" da0="0" da1="0" da2="0" />
   <bond from="1" to="2" da0="0" da1="0" da2="0" />
   <bond from="1" to="0" da0="1" da1="0" da2="0" />
   <bond from="2" to="0" da0="0" da1="1" da2="0" />
   <bond from="2" to="1" da0="-1" da1="1" da2="0" />
</unitcell>
```

```
<model name="kagome-DM">
   <interaction parameter="j" from="0,0,0,1" to="0,0,0,0" ↩
      type="heisenberg" />
   <interaction parameter="d" from="0,0,0,1" to="0,0,0,0" ↩
      type="xy" />
   <interaction parameter="d" from="0,0,0,1" to="0,0,0,0" ↩
      type="-yx" />
   <interaction parameter="j" from="0,0,0,0" to="0,0,0,2" ↩
      type="heisenberg" />
   <interaction parameter="d" from="0,0,0,0" to="0,0,0,2" ↩
      type="xy" />
   <interaction parameter="d" from="0,0,0,0" to="0,0,0,2" ↩
      type="-yx" />
   <interaction parameter="j" from="-1,0,0,1" to="0,0,0,0" ↩
      type="heisenberg" />
   <interaction parameter="d" from="-1,0,0,1" to="0,0,0,0" ↩
      type="xy" />
   <interaction parameter="d" from="-1,0,0,1" to="0,0,0,0" ↩
      type="-yx" />
   <interaction parameter="j" from="0,0,0,0" to="0,-1,0,2" ↩
      type="heisenberg" />
   <interaction parameter="d" from="0,0,0,0" to="0,-1,0,2" ↩
      type="xy" />
   <interaction parameter="d" from="0,0,0,0" to="0,-1,0,2" ↩
      type="-yx" />
   <interaction parameter="j" from="0,0,0,2" to="0,0,0,1" ↩
      type="heisenberg" />
   <interaction parameter="d" from="0,0,0,2" to="0,0,0,1" ↩
      type="xy" />
   <interaction parameter="d" from="0,0,0,2" to="0,0,0,1" ↩
      type="-yx" />
   <interaction parameter="j" from="0,-1,0,2" to="-1,0,0,1" ↩
      type="heisenberg" />
   <interaction parameter="d" from="0,-1,0,2" to="-1,0,0,1" ↩
      type="xy" />
   <interaction parameter="d" from="0,-1,0,2" to="-1,0,0,1" ↩
      type="-yx" />
</model>
```

Both definitions are included in the default installation of the SpinParser software. They are stored within in the files `$SPINPARSER/res/lattices.xml` and `$SPINPARSER/res/models.xml`, respectively, where `$SPINPARSER` is the root directory of the SpinParser installation. While the lattice definition in the code listing above is straightforward, the definition of the spin interactions is more involved. The Heisenberg interactions are defined in close analogy to the previous example on the cubic lattice in Sec. 6.1. The definition of DM interactions requires more attention: Since the DM interactions break the lattice bond inversion symmetry and carry a sense of direction, the order of sites in the `from` and `to` attributes becomes relevant, see the definition of exchange terms in Eq. (47); for the six nearest neighbor interactions per unit cell we adopt a convention of bond orientations as illustrated in Fig. 8b. Furthermore, for the out-of-plane DM vectors in our example, each interaction contains two terms, i.e., the expression $h_{ij} = \mathbf{D}_{ij} \cdot (\mathbf{S}_i \times \mathbf{S}_j)$ for the DM vector pointing out of plane, $\mathbf{D}_{ij} = (0, 0, 1)$, amounts to $h_{ij} = S_i^x S_j^y - S_i^y S_j^x$. Due

to the in general arbitrary nature of the DM vector orientation, a shorthand notation for DM interactions does not exist in SpinParser. Instead, on every bond, the two terms of the interaction (`type="xy"` and `type="-yx"`) are defined separately.

With the model definition set up, we now prepare a task file `$SPINPARSER/examples/kagome-DM.xml` as described in Sec. 5.1:

```
<task>
    <parameters>
        <frequency discretization="exponential">
            <min>0.005</min>
            <max>50.0</max>
            <count>64</count>
        </frequency>
        <cutoff discretization="exponential">
            <max>50.0</max>
            <min>0.01</min>
            <step>0.98</step>
        </cutoff>
        <lattice name="kagome" range="7"/>
        <model name="kagome-DM" symmetry="TRI">
            <j>1.000000</j>
            <d>0.000000</d>
        </model>
    </parameters>
    <measurements>
        <measurement name="correlation"/>
    </measurements>
</task>
```

Note that due to the broken spin rotation symmetry of our model, we need to employ the most general of the numerical backends, which is suitable for general models of two-spin interactions as captured by the Hamiltonian Eq. (1). The choice of the numerical backend is specified in the attribute `symmetry="TRI"` in the code listing above. Further, we defined the coupling constants to be $J = 1.0$ and $D = 0.0$; they are, of course, subject to change as we explore the parameter space of the model. Owed to the reduced symmetry of the model, the calculation is significantly more complex than in the previous example. The calculation takes approximately 2,500 core hours on Intel Xeon Phi 7250-F Knights Landing processors (68 cores, 1.4 GHz) when parallelized across 4 distributed memory nodes with a total of 272 physical CPU cores.

We now extract the spin correlation measurements and the spin structure factor. In the output file `$SPINPARSER/examples/kagome-DM.obs`, with our choice of the numerical backend, the lattice site-resolved spin correlations $\chi_{ij}^{zz,\Lambda}$ are stored in the HDF5 datasets `/TRICorZZ/data/measurement_*/data`, as described in Sec. 5.4. Similarly, we gather the two remaining components $\chi_{ij}^{xx,\Lambda}$ and $\chi_{ij}^{yy,\Lambda}$ to compute the structure factor $\chi^{\Lambda}(\mathbf{k})$. Unlike in the previous example of the cubic lattice model, where the ground state would always exhibit magnetic order regardless of the choice of parameters, the Heisenberg-DM model on the kagome lattice is expected to host a spin liquid ground state in the vicinity of the Heisenberg limit $D = 0.0$. The quantum spin liquid ground state, which preserves the spin rotation symmetry of the system, manifests in a smooth flow of the spin correlations down to the lowest cutoff, as shown for the peak susceptibility $\chi_{\max}^{\Lambda}$ at $D = 0.0$ in Fig. 10a. At the same time, the momentum space structure of the correlations remains mostly featureless, exhibiting only broad maxima around the edge of the extended Brillouin zone (inset of Fig. 10a); this is a well known aspect of the kagome Heisenberg antiferromagnet, which has been observed in a number of pf-FRG studies [26, 28, 47, 48].

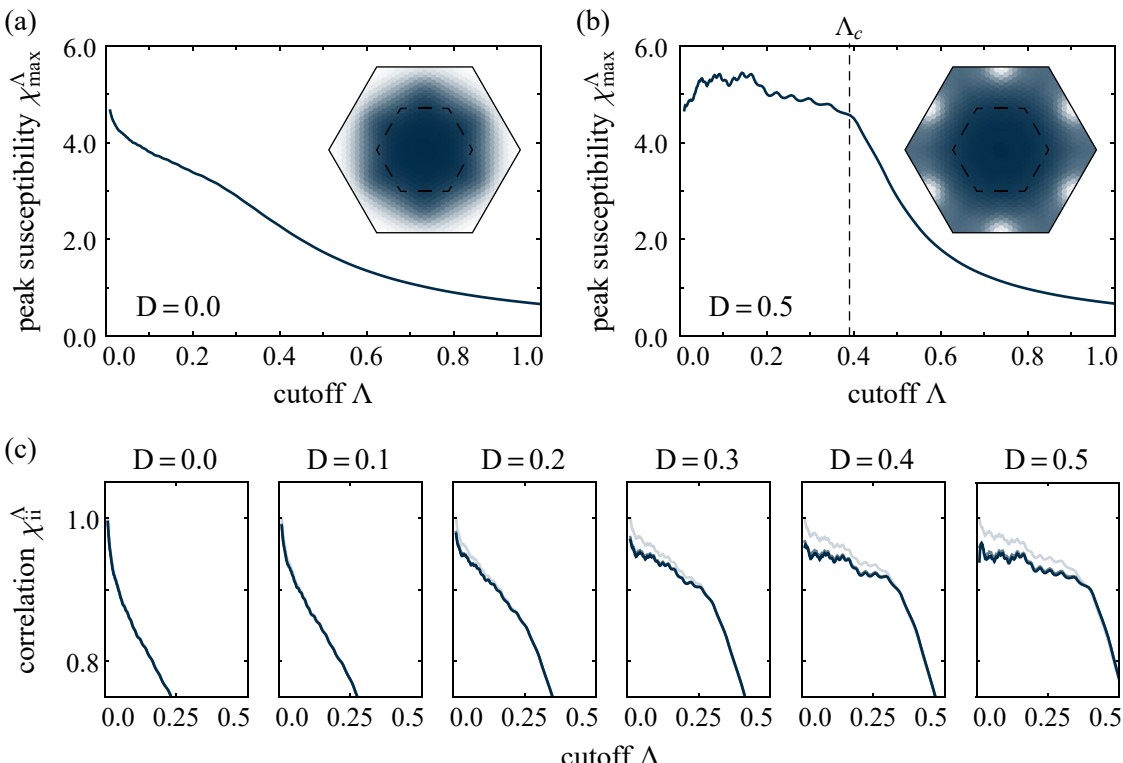

Figure 10: **Renormalization group flow** of the two-spin correlations in the Heisenberg-DM model on the kagome lattice. (a) Flow of the peak susceptibility at vanishing DM interactions $D = 0$. (b) Flow of the peak susceptibility at $D = 0.5$, indicating a flow breakdown at $\Lambda_c = 0.39$. The insets of panels (a) and (b) show the spin correlations across the entire extended Brillouin zone, plotted at $\Lambda = 0$ and $\Lambda = \Lambda_c$, respectively. The color code is normalized independently, ranging from low intensity (dark blue) to high intensity (white). (c) Scaling of the onsite correlation $\chi_{ii}^{\Lambda}$ with the lattice truncation range $L$. Data shown is for $L = 3$ (lightest color), $L = 5$, and $L = 7$ (opaque color).

Increasing the DM interaction strength $D$, on the other hand, is expected to eventually give rise to magnetic order. Indeed, setting $D = 0.5$, we observe a flow breakdown at finite $\Lambda_c = 0.39$ indicative of spontaneous symmetry breaking. The associated structure factor at the critical scale exhibits sharply localized peaks at the boundary of the extended Brillouin zone, see Fig. 10b.

Naturally, there must exist a phase boundary between the spin liquid phase around $D = 0.0$ and the magnetically ordered phase near $D = 0.5$. How can we localize such boundary? The flow breakdown in pf-FRG calculations for intricate magnetic order can sometimes be subtle – in the present example (Fig. 10b), the breakdown is much less pronounced as e.g. in the Heisenberg antiferromagnet on the cubic lattice (Fig. 9a). In the past, it has proven useful to assess the scaling of the onsite spin correlations $\chi_{ii}^{\Lambda}$ with the lattice truncation range $L$ [37, 46]. The expression for the onsite spin correlations contains contributions from all vertex functions within the lattice truncation range. If correlations in the system are only short-ranged, the result quickly converges in the truncation range $L$. If, on the other hand, correlations become long-ranged – i.e., for cutoff values $\Lambda < \Lambda_c$ in systems which exhibit magnetic order – the convergence behavior suddenly changes, and convergence in the truncation range $L$ becomes slower. This is illustrated in Fig. 10c, where we plot the onsite spin correlations for several values of the DM interaction strength $D$. For interactions $D \leq 0.1$ the result is fully converged for all values of the truncation range $L = 3, 5, 7$. In contrast, at $D \geq 0.2$, a change in the scaling behavior becomes visible below some critical scale $\Lambda_c$: At $\Lambda < \Lambda_c$ the curves begin to differ for different lattice truncation ranges. We can

thus conclude that the phase transition lies between $D = 0.1$ and $D = 0.2$. In fact, it has been estimated before by pf-FRG methods [47, 48] and exact diagonalization [69] that the transition point lies near $D \approx 0.1$.

# 7  Conclusions

We have discussed technical aspects of the pf-FRG algorithm as implemented in the SpinParser software. With the SpinParser software being the first publicly available implementation of the pf-FRG algorithm which offers support for the numerical solution of the general spin Hamiltonian given in Eq. (1), it marks a significant step in making the study of a broad class of two- and three-dimensional quantum spin models – which are often notoriously difficult to treat with established numerical techniques – more accessible. We explained how custom lattice spin models, as well as other relevant parameters for the computation, can be defined in a plain-text input format, and we demonstrated the use of the SpinParser software on the basis of two fully worked out examples.

We have further demonstrated that, despite providing a high-level interface for the specification of the lattice spin model of interest, the SpinParser software with its underlying numerical architecture remains highly efficient and can be parallelized across a large number of shared memory and/or distributed memory compute nodes. We illustrated the latter by showing benchmark results for calculations on up to 1088 CPU cores (16 distributed memory compute nodes), which indicated only a small parallelization overhead. The high efficiency of the pf-FRG computations is achieved with the help of an automatic symmetry analysis of the lattice spin model, which is built into the SpinParser software. The findings of the symmetry analysis are leveraged to construct a maximally reduced parametrization of the pseudofermionic vertex functions, which are at the center of any pf-FRG calculation. We provided benchmark calculations which illustrate the scaling of the computing time as a function of the lattice truncation range and the frequency discretization – the two main parameters with regard to the numerical accuracy of the solution – of the vertex functions.

While the SpinParser software can offer an 'out-of-the-box' experience for the solution of many problems of current interest in quantum magnetism, it is not complete: Research on the pf-FRG algorithm, its methodological advancement, and its numerical implementation is still ongoing, continuously generating new concepts for refinements or extensions of the method. One intriguing aspect for future revisions of the code would be the implementation of the recently proposed "multiloop" extension of the pf-FRG algorithm [56, 70, 71]. In combination with a more careful treatment of the frequency dependence of the vertex functions [72], it could add a new level of quantitative control to the pf-FRG.

# Acknowledgements

The author thanks Dominik Kiese and Simon Trebst for work on related projects which stimulated the development of the code, as well as Daniel Rohe and Edoardo Di Napoli from the Juelich Supercomputing Centre for consulting on aspects of the numerical implementation. This work was partially supported by the DFG within the CRC 1238 (project C02). Numerical simulations were performed on the JURECA Booster at the Forschungszentrum Juelich.

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
