# Peer review of "The SpinParser software for pseudofermion functional renormalization group calculations on quantum magnets"

_SciPost Physics Codebases_

## Round 1 · Referee Report · Anonymous (Referee 1) · 2021-12-1

Strengths

1- The Spin Parser open source software is extremely versatile as it is able to address any given two-spin interaction Hamiltonian defined on generic two- and three-dimensional lattices which can be constructed in custom fashion. This is in addition to the already provided constructions for standard models and lattices.

2- The field of frustrated quantum magnetism poised with the arrival of materials and models which promise to host quantum spin liquid behavior especially in three-dimensions. The availability of this software opens the avenues to the entire community of quantum magnetism researchers (theoreticians and experimentalists alike) to access the powerful pf-FRG framework which has established itself in being able to provide valuable insights into spin models in 3D, both isotropic and the highly involved spin anisotropic Hamiltonians, which in 3D remain out-of-reach of most numerical approaches. In being able to compute static susceptibilities the method reaches out to experimentalists and will prove to be an extremely valuable tool for neutron scatterers.

3- The code is heavily optimized in not only exploiting both real space and spin space symmetries in conjunction, but also in providing an excellent performance when deployed on shared and/or distributed memory compute nodes.

4- The manuscript documents all aspects of the open source software in a very systematic and explicit fashion. This makes it user friendly. It also provides a nice and pragmatic introduction to the pf-FRG framework, which conveys and appropriately highlights all the essential points and quickly introduces one to a working knowledge of the same.

Weaknesses

1- The Spin Parser software is designed in such a way that its working is restricted to lattices in which all sites a equivalent, i.e., an arbitrary site can be mapped to a given "origin" by employing translations and point group operations. This unfortunately excludes the family of non-Archimedean lattices, which contain n-inequivalent sites. Given the fact that some of them, e.g., the square-kagome lattice, and bi-trillium lattices have recently come into limelight experimentally in showing evidence for quantum spin liquid behavior [see https://www.nature.com/articles/s41467-020-17235-z, and https://journals.aps.org/prl/abstract/10.1103/PhysRevLett.127.157204], it may be worthwhile to consider extending the Spin Parser software such that its able to address non-Archimedean geometries.
I understand that this will increase the computational effort as the spin-spin correlations in real space would now have to computed from the "n" different/inequivalent origin sites, and also the self-energy will acquire an additional index, however, from a symmetry reduction point of view this does not add much complication. iF I understand the current implementation, this task seems feasible without much effort. The increase in computational effort is also moderate as most cases of interest involve, e.g., 2 distinct origins, and in general this aspect can be parallelized.

2- In the case of three dimensional lattices with a basis, like the pyrochlore lattice, it is very important that the manuscript/documentation mention that the Hamiltonian in pf-FRG is defined in the local frame of reference/basis, and not the global one. This is so because when producing the susceptibility plots to compare with neutron scattering data, one has to carry out a transformation of coordinates to produce the data in the global frame which can then be compared to neutron scattering profiles. The current data output given by the code is in the local frame of reference.

3- Since this manuscript will serve to be a very valuable tool to experimental groups also, it may prove helpful if the code output is slightly re-designed to produce the susceptibility data in both the spin-flip and non spin-flip channels. From the way the current output files are written, it proves a bit cumbersome to do this post-processing of data.

4- The current implementation is restricted to spin S=1/2 moments for arbitrary two body Hamiltonians, and works for spin S>1/2 only for Heisenberg models. Maybe I missed some aspect, but I was curious what issues in the implementation would arise if one has S>1/2 moments interacting with a generic anisotropic interaction.

Report

I would be grateful if the author could please take into consideration the remarks I put forth in the Weaknesses section. I would not classify them strictly as weaknesses or shortcomings but in my assessment, these are suggestions which are likely to aid towards making the Spin Parser of wider utility, usefulness, and reach to the broad community and thus enhance its impact.

I have gone through the Acceptance Criteria of "SciPost Physics Codebases" and in my opinion, all these criteria are very well met.

Overall, I would strongly recommend the publication of this manuscript and software in SciPost Physics Codebases. However, it would be nice If the author could at least incorporate some of the points mentioned in the weaknesses section, and re-submit a revised manuscript and code. If these seem not feasible, it would be great to have the author's response to these recommendations explaining what issues arise.

Requested changes

Points 1-4 in the Weaknesses section.

  • validity: high
  • significance: high
  • originality: good
  • clarity: top
  • formatting: excellent
  • grammar: excellent

Author:  Finn Lasse Buessen  on 2022-02-22  [id 2241]

(in reply to Report 1 on 2021-12-01)
Category:
answer to question

Thank you for your thoughtful feedback and for your suggestions for further improvement of the manuscript. Detailed answers to each of your comments are provided below.

1- The restriction to Archimedean lattices indeed excludes application to some spin models which may be of interest. As you correctly point out, an extension of the pf-FRG algorithm to cover also non-Archimedean lattices is straight-forward on the conceptual level. However, such an extension would also imply significant changes to the numerical backend of the software since the structure of the flow equations themselves would change significantly; the changes would require extensive code testing and validation of the added use cases, while ensuring that the performance for Archimedean lattices is not negatively impacted. Since such an extension would not affect the front-end of the SpinParser software (i.e., the format in which lattice spin models and computational tasks are defined), I believe that it is appropriate to publish the software in its current state and pursue the extension to non-Archimedean lattices in a future revision of the code; already in its current state, the software is applicable to a plethora of spin models of current interest and can be of great use to the community.

2- Thank you for catching this potential source of confusion. I have added a paragraph in Section 5.3 (“Definition of spin interactions”) to clarify the definition of spin operators and their frame of reference. I further added brief remarks below Eq. (4) (i.e., the definition of the two-spin correlation function) and in the fourth paragraph of Section 5.4 (“Output data”) to mention this subtlety. Similarly, I have added a paragraph in the “Prepare a task file” section of the quick start guide in the code documentation.

3- Thank you for bringing up the question about the storage format of output data and potential usability issues. The output files generated by SpinParser contain the full set of real-space resolved two-spin correlations, which indeed require some postprocessing to relate to typically probed physical quantities. To simplify postprocessing, a set of Python tools that provides convenient access to the correlation functions both in real space and in momentum space (via Fourier transformation) is included with the SpinParser software. In order to make the Python tools more accessible, I have added a separate user documentation for the Python tools [https://fbuessen.github.io/SpinParser/doc-python], which is also referenced in the README. Furthermore, to aid experimentalists in computing more complicated correlation functions like the expressions for spin-flip and non spin-flip scattering, I have added an extra paragraph and example code in the “Evaluate SpinParser output and measurements” section of the README, which illustrates a convenient way to compute such expressions in a few lines of Python code.

4- The extension of the pf-FRG algorithm to arbitrary spin length S has originally been proposed in [PRB 96, 045144 (2017)] for Heisenberg models. The spin-S generalization contains some pitfalls that make it less suited for spin models with lowered symmetry. Like the original pf-FRG scheme, the extension is based on a representation of spin operators in terms of pseudofermions: The S>1/2 operators are first decomposed into multiple replicas of S=1/2 operators, which are then represented by multiple flavors of pseudofermions. However, for this representation to be faithful, all S=1/2 replicas (all pseudofermion flavors) must locally align in a ferromagnetic fashion. As demonstrated in [PRB 96, 045144 (2017)], this tends to happen naturally for Heisenberg models; the alignment constraint can be verified or energetically enforced by adding an on-site coupling term -A*S^2, which would energetically punish the unphysical lower-spin sectors of the (replica) pseudofermion representation for some A>0. The constant A needs to be chosen large enough to enforce the constraint, yet it cannot be chosen too large, since that would interfere with the ability to (accurately) numerically resolve the physical exchange couplings of interest. The question of whether meaningful physical results can be produced in a similar way for spin models with non-Heisenberg interactions, which would often favor non-collinear alignment of spins, has not yet been studied. Since the focus of the SpinParser software is to make established and well understood pf-FRG techniques readily available, a formal spin-S generalization for non-Heisenberg models is not implemented.

---

## Round 1 · Referee Report · Johannes Hauschild (Referee 2) · 2022-1-10

Strengths

1-The library provides a flexible implementation of the pf-FRG algorithm capable of dealing with quite generic lattices and interactions.
At the same time, the library is made efficient by exploiting symmetries and providing MPI parallelization.
The black-box like interface makes it easy to run the code even without detailed knowledge of all the details of pf-FRG.

2-Both the manuscript and the user guide of the library are well written with examples explaining the typical use cases.

Weaknesses

1-In the introduction, it is explained that pfFRG becomes exact in the limit of large spin S. However, as far as I understand, only Spin 1/2 is supported by the current implementation. Please clarify this in the manuscript, and possibly also in the user guide/ReadMe.
(A generalization to higher spin would be nice but is certainly not a requirement for a publication.)

Report

The library and manuscript easily meet all the acceptance criteria.
As the first open implementation of pf-FRG, it will certainly be useful to the community.
I therefore recommend a publication after a very minor revision.

Requested changes

See Weakness section above.

Typos:
- duplicated "typically" at end of page 20
- Reference to Sec. 6 instead of 5 in the outline at end of Sec. 1.

  • validity: top
  • significance: top
  • originality: top
  • clarity: top
  • formatting: perfect
  • grammar: perfect

Author:  Finn Lasse Buessen  on 2022-02-22  [id 2240]

(in reply to Report 2 by Johannes Hauschild on 2022-01-10)
Category:
answer to question

Thank you for your careful reading of the manuscript and for the very positive assessment of the work, as well as for spotting the typos.

Regarding the question of whether calculations for generalized spin-S models are possible: A generalization of pf-FRG calculations to spin-S Heisenberg models has been proposed in [PRB 96, 045144 (2017)] (see also the response to “Anonymous Report 1 on 2021-12-1”). This generalization is implemented in the SpinParser code, but it had not been documented appropriately before. In the revised manuscript, I have added a paragraph in Section 5.1 (“Structure of a task file”) which provides instructions on how to perform such generalized spin-S calculations for Heisenberg models. Furthermore, I have updated the example that is showcased in the “Quick start” section of the README to include a discussion of the spin-S generalization.

---

## Editorial Decision

resubmitted